# KERNEL BINARY OPTIMIZER (KBOP): LATENT-FREE OPTIMIZER FOR BINARY NEURAL NETWORKS

## ABSTRACT

Binary Neural Networks (BNNs) are receiving growing interest for enabling energy-intensive deep learning on resource-limited edge devices. Traditionally, the training methods of such models rely on the minimizing the quantization error in forward propagation and approximation of sign function in full precision models. However, such methods do not use the nature of training in the space of binary weights. To address this issue, we propose a latent-free method called Kernel Binary Optimizer (KBOP) to realize binary deep learning. Proposal is largely based on three new major technical solutions: a sign-changing rule that reverses the sign of a binary weight in accordance with the value of the gradient calculated at the binary point, the rigidity of this rule is regulated by the learning rate (LR); a learnable scaling factor that allows to partially integrate the full precision nature of the input into the binary optimization space with a LR different from the sign-changing rule; BNN Initialization (BNN Init) procedure allowing start binary learning more stable. The novelty of KBOP lies not only in its binary optimization approach but also in its seamless integration into existing architectures, such as convolutional or transformer networks. To demonstrate this, we compare our approach to the state-of-the-art (SoTA) BNN training methods for super-resolution task and large language models, achieving a significant inference performance increase. Experimental results indicate that our method covers, on average, 30.82% of the gap between the existing SoTA BNN models results and full-precision model performance. We also provide a theoretical proof of the convergence of KBOP and provide a theoretical justification for BNN Init.

## 1 INTRODUCTION

Deep neural networks (DNNs) have achieved SoTA results across various domains (Tsirmpas et al., 2024; Archana & Jeevaraj, 2024), but their growing computational and memory demands challenge deployment on resource-limited devices. Quantization addresses this by lowering the precision of weights and activations, reducing storage and inference costs. Binarization, the most extreme form, uses 1-bit parameters, enabling major speedups (Rastegari et al., 2016). Binary neural networks (BNNs) are well-suited for edge and mobile devices due to their efficiency, though they often lag behind full-precision models in accuracy — pushing the ongoing research to close this gap.

Research on BNN training (Yuan & Agaian, 2023) is typically divided into Post-Training Quantization (PTQ) and Quantization-Aware Training (QAT). PTQ quantizes a pretrained FP model but suffers from accuracy loss due to missing weight adaptation (e.g., Li et al. (2021); Hubara et al. (2021)). QAT performs quantization during training, allowing the model to adapt to binarization and converge more stably. Therefore our work focuses solely on QAT, it can be categorized in the following way (Yuan & Agaian, 2023): *Loss Function Modification* (Xu et al., 2023; Liu et al., 2020) – add regularization terms or distribution-based losses to guide binarization; *Quantization Error Minimization* (Qin et al., 2020; Falkena et al., 2023; Rastegari et al., 2016) – reducing information loss during quantization; *Network Topology* (Zhang et al., 2021; Liu et al., 2020; 2018) – designing or modifying architectures to support binarization; *Training strategies* (Liu et al., 2020; Qiu et al., 2022; Liu et al., 2018) – use of specific training techniques to improve performance; *Gradient Approximation* (Kim et al., 2020; Qiu et al., 2022; Qin et al., 2020) – surrogate gradients to handle the zero-gradient problem.

Despite recent progress, training stable and accurate binary models remains challenging (Yuan & Agaian, 2023). A key issue is the use of full-precision latent weights during training, which creates a mismatch between real-valued gradients and binary forward passes. Optimizers like Adam or SGD assume continuity, but binarization breaks this assumption, leading to inaccurate updates. Existing solutions such as gradient approximations or loss tweaks do not usually fully address the gap between latent and binary weights, especially when weight values are small, causing unstable training and frequent bit flips. This highlights the need for training methods tailored to binary constraints, directly optimizing +1/-1 weights.

Binarization challenges become critical in tasks demanding fine-grained tuning to match full-precision (FP) performance (Yuan & Agaian, 2023), especially in complex domains like computer vision and natural language processing (NLP). We investigate BNNs in three representative scenarios: image super-resolution (SR), large language models (LLMs) and image classification. SR, sensitive to quantization artifacts, is evaluated using EDSR (Lim et al., 2017b) on DIV2K (Vargas et al., 2024), with benchmarks like Set5/Set14 and metrics such as PSNR and SSIM. For LLMs, we test GPT-2 on IMDB (Maas et al., 2011) and WikiText-2 (Merity et al., 2016), using perplexity and accuracy metrics respectively. To assess binarization impact, we also track $\mu_q$ (quality gap), $\mu_m$ (memory usage), and $\mu_p$ (training stability). For comparing with the SoTA methods we choose ReActNet and LAB as they show the highest performance on ResNet-18 for ImageNet. Furthermore, we add IR-Net which demonstrates performance close to SoTA, and BBCU as the SoTA approach in SR.

We propose KBOP, a novel latent-free BNN training algorithm that tackles binary optimization via a novel sign-flip rule based on accumulated binary-point gradients, and uses its own learning rate (KBOP LR) to control the number of binary weights being modified. It also makes use of a learnable scaling factor with a separate LR to reduce quantization error, and a novel BNN-specific initialization to stabilize training (BNN Init). These components together provide gradient-aware training method that can be easily integrated into existing architectures without structural changes. We also provide a theoretical proof of the convergence of KBOP and a theoretical justification for BNN Init. For benchmarking we choose one or more leading approaches in each of the five training BNN training method categories (Yuan & Agaian, 2023).

## 2  RELATED WORK

Gradient approximation techniques for BNN training were first introduced by Courbariaux et al. (2015). BiReal-Net (Liu et al., 2018) employs a piecewise polynomial approximation for the sign function. IR-Net (Qin et al., 2020) sees binarization through the view of information flow and introduce weight standardization along with a dynamic gradient estimator to minimize errors from gradient approximation.

To reduce binarization-induced information loss, various methods aim to minimize quantization error. AdaBin (Tu et al., 2022) adaptively selects optimal binary sets per layer, while XNOR-Net (Rastegari et al., 2016) uses channel-wise scaling. LAB (Falkena et al., 2023) replaces the sign function with a learnable binarizer to address representational limits. BiPer (Vargas et al., 2024) leverages periodic functions for smoother forward and backward passes. Orecchia et al. (2025) apply variational inference for optimization, and ReActNet (Liu et al., 2020) improves training through distributional loss and activation reshaping.

Other works of particular relevance to this paper are related to latent-free BNN training methods. DPCD Xiong (2022) defines a sign-change rule, which move weight from positive set to negative and vice versa based on the gradient calculated in binary points. An implementation of a similar idea for neural networks can be found in Binary Optimizer (Bop) (Helwegen et al., 2019b). It handles binary weights and accumulates gradients to decide when to flip a weight's sign. Based on Bop, Suarez-Ramirez et al. (2021) presented Bop2ndOrder, suggesting the use of the second moment estimation, similar to the one used in Adam (Kingma, 2014).

In the field of image super-resolution, E2FIF (Song et al., 2023) utilizes an end-to-end full-precision information flow in BNNs to improve expressiveness during forward propagation and the precision of backpropagated gradients. BBCU (Xia et al., 2022) introduces a basic binary convolutional unit that eliminates batch normalization (BN) in the binarized model.

For comprehensive evaluation of the proposed algorithm, we curated a set of baseline methods exhibiting SoTA performance, ensuring representation across the basic categories of BNN methodologies (Yuan & Agaian, 2023). The method selection criteria prioritized empirical effectiveness on challenging benchmark tasks: image classification (CLF) using ResNet-18 (He et al., 2016) and image super-resolution (SR) using EDSR --— as presented in table 1 and table 2, respectively. Superior performance on these architectures indicates the robustness and generalizability of BNN optimization strategies, thereby validating their suitability for comparison. Specifically, we incorporate ReAct-Net (Liu et al., 2020), LAB (Falkena et al., 2023), and IR-Net (Qin et al., 2020), each demonstrating strong results on the ImageNet dataset with ResNet-18; BBCU (Xia et al., 2022), as it is efficient for SR tasks; and the original BNN framework of Courbariaux et al. (2016), employed as a foundational reference to assess theoretical advancements.

Table 1: Accuracy comparison of SoTA BNN training methods on ImageNet dataset and ResNet18 architecture.

| Method | Top-1 acc. (%) | Top-5 acc. (%) |
|---|---|---|
| FP | 69.6 | 89.2 |
| BNN | 42.2 | – |
| IR-Net | 58.1 | 80.0 |
| BinaryDuo | 60.4 | 82.3 |
| BiPer | 61.4 | 83.14 |
| ReBNN | 61.6 | 83.4 |
| VISPA | 62.1 | 83.4 |
| AdaBin | 63.1 | 84.3 |
| LAB | **64.2** | **85.0** |

Table 2: Performance comparison of SoTA BNN training methods on image SR task (EDSR architecture) and Set5, Set14 datasets.

| Topology | Method | Set5 | | Set14 | |
|---|---|---|---|---|---|
| | | PSNR | SSIM | PSNR | SSIM |
| EDSR (x4 scale) | FP | 32.46 | 0.897 | 28.80 | 0.787 |
| | BNN | 17.53 | 0.188 | 17.51 | 0.160 |
| | E2FIF | **31.91** | **0.890** | **28.29** | **0.775** |

## 3 KERNEL BINARY OPTIMIZER

This section introduces a novel latent-free BNN traninng algorithm, KBOP. We begin by discussing the binarization of weights and activations during forward and backward propagation in the first subsection. Then we discuss a general way to initialize parameters of KBOP. We also discuss the rationale behind KBOP's update rule.

**Forward propagation.** The output of a CNN layer in DNNs with weights vector $\boldsymbol{w} \in \mathbb{R}^d$ and previous layer activations $\boldsymbol{a} \in \mathbb{R}^p$ can be expressed by

$$z = \boldsymbol{w}\boldsymbol{a}.$$

We formulate the output of a layer with binary weights $\boldsymbol{w} \in \{-1, 1\}^d$ and FP activations $\boldsymbol{a}$ as

$$z = \alpha \boldsymbol{w} \operatorname{sign}(\boldsymbol{a}),$$

where $\alpha \in \mathbb{R}$ is a learnable layer-wise scaling factor, $\operatorname{sign}(\boldsymbol{x})$ denotes element-wise $\operatorname{sign}$ function applied to vector $\boldsymbol{x}$. Note that $\alpha$ can be negative.

**Backward propagation.** Since the derivative of $\operatorname{sign}$ function is zero almost everywhere, its straightforward usage is incompatible with backward propagation algorithm. This leads to the introduction of gradient approximation techniques for BNN training that can be expressed as

$$\frac{\partial L}{\partial \boldsymbol{a}} = \frac{\partial L}{\partial Q_a(\boldsymbol{a})} \frac{\partial Q_a(\boldsymbol{a})}{\partial \boldsymbol{a}} \approx \frac{\partial L}{\partial Q_a(\boldsymbol{a})} h'(\boldsymbol{a}),$$

$$Q_a(\boldsymbol{a}) = \operatorname{sign}(\boldsymbol{a}),$$

where $L : \mathbb{R}^d \to \mathbb{R}$ denotes the loss function, $Q_a$ represent the activation binarization function, correspondingly, and $h$ is the continuously differentiable approximation of $Q_a$.

Denote the loss of a BNN as $\overline{L} : \mathbb{R}^d \times \mathbb{R} : (\boldsymbol{w}, \alpha) \mapsto L(\alpha \boldsymbol{w})$. Then, gradient of $\overline{L}$ w.r.t. weights $w_i$ and scaling factors $\alpha$ are computed using common backpropagation algorithm:

$$\frac{\partial \overline{L}}{\partial w_i} = \alpha \partial_i L|_{\alpha \boldsymbol{w}}, \ i = 1, \ldots, d,$$

where $\partial_i L$ is partial derivative of $L$ w.r.t. to its $i$-th variable.

Various approximation methods have been proposed in prior works, which can be categorized based on whether they evolve over iterations or epochs: approximations (Courbariaux et al., 2015; Liu et al., 2018) and dynamic approximations (Qin et al., 2020; Lu et al., 2024; Cai et al., 2024). In our study, we use baseline's approximations of activations.

KBOP works in binary space, so the update rule from SGD-like algorithms (shift along the direction of the anti-gradients) for binary weights does not works and needs to be modified. Other latent-free optimizers, such as Bop (Helwegen et al., 2019b) and Bop2Order (Suarez-Ramirez et al., 2021) uses constant threshold rule: binary weight change sign if corresponding moment of gradient greater than some constant number. This leads either to getting stuck with a large threshold or to overfitting with a small one. To address this issue, the rule for updating the sign of the weights should take into account the local structure of the loss near the binarization point, in other words, the rule should be entirely determined by local characteristics of the loss — such as gradients, Lipschitz constants, etc. In this work, we propose the **Kernel Rule**, which is local by design.

### 3.1 KERNEL RULE

For each layer let $\mathbb{S}$ be the set of weights indices that should change sign at the current iteration of learning, and $v_i$ – components of first moment of gradients:

$$v_i^{k+1} = \beta v_i^k + (1 - \beta) \frac{\partial \overline{L}(\boldsymbol{w}^k, \alpha_k)}{\partial w_i}$$

where $k$ denotes training iteration number. We define $S$ as follows:

$$\mathbb{S} = \{i \mid w_i v_i > 0, \ \lambda \, ||v_i| - l| > \sigma\} . \tag{1}$$

Here, $l$ and $\sigma$ denote the mean and the standard deviation of the absolute values of the components of $v$ respectively; and $\lambda$ is the LR for KBOP:

$$l = mean\left(|v_i|\right), \tag{2}$$

$$\sigma = std\left(|v_i|\right). \tag{3}$$

Thus, $1/\lambda$ corresponds to the width of the confidence interval for $|v_i|$, measured in standard deviations. Using Chebyshev's inequality, one can derive that under such a rule, the probability that a given weight changes its sign is less than $\lambda^2$, which provides an intuitive interpretation of the LR for KBOP.

### 3.2 BNN INITIALIZATION.

The main idea of BNN Init is to improve the numerical stability of the network by choosing an appropriate initial scaling factor. Let $n$ be the number of weights in current layer. We choose initial value of $\alpha = \sqrt{2/n}$. The detailed derivation of the chosen initial scaling factor is presented in Section 4.1. The corresponding pseudocode is presented in Algorithm 1.

---
**Algorithm 1** BNN Init
---
**Input**: Uninitialized binary network with binary weights $w$ and scaling factors $\alpha$.

    Initialize $w$ using a Bernoulli distribution with $p = 0.5$.

    Initialize $\alpha$ with the value $\sqrt{2/n}$, where $n$ is the number of weights in the layer.

---

## 3.3 ALGORITHM AND NOVELTY

The pseudocode outlining the proposed method is presented in Algorithm 2. The proposed KBOP method introduces several key novelties:

1. **Momentum-based update rule without sign constraint.** Unlike previous approaches that apply a sign function to gradients or use fixed thresholds, KBOP directly uses the first momentum of the floating-point gradients at binary weights to define update candidates. This preserves information about the loss landscape.

2. **Kernel rule with confidence intervals.** The update rule is based on a local statistical characteristic — a confidence interval centered at the average gradient magnitude, with a width controlled by a KBOP-specific LR $\lambda$. This replaces fixed thresholds and allows the optimizer to adapt to the variability of the gradient signal.

3. **KBOP LR scheduler.** LR $\lambda$ controlling the width of the confidence interval is scheduled over time (e.g., via Step, Cosine Annealing, or Linear decay), enabling finer control over the aggressiveness of binary weight updates during training.

4. **Binary-specific initialization (BNN Init).** A novel initialization strategy for the scaling factor $\alpha$ and binary weights is introduced, inspired by Kaiming initialization, improving the numerical stability and convergence of BNN training.

These features make KBOP more robust to overfitting, less prone to premature convergence, and applicable across a wide range of binary neural network architectures.

---

**Algorithm 2** BNN training using KBOP

---

**Input:** Loss function $\overline{L}$, smooth approximation $h$ of sign, momentum coefficient $\beta$, KBOP LR schedule $\lambda_k$ (e.g., Step LR, Cosine Annealing LR)

**Binary-specific initialization (BNN Init):** Initialize binary weights $\boldsymbol{w}^0 \in \{-1, 1\}^d$ and scaling factor $\alpha_0 \in \mathbb{R}^+$ using variance-aware heuristic                            *// Novelty 4*

**Trainable scaling factors:** Use LR $\mu_k$ for $\alpha_k$ updates

  **while** not converged or maximum iterations not reached **do**

      **Forward propagation**

    • Compute binarized activations: $\tilde{\boldsymbol{a}} = \text{sign}(\boldsymbol{a})$

    • Compute output: $z = \alpha_k \boldsymbol{w}^k \tilde{\boldsymbol{a}}$

      **Backpropagation**

    • Compute gradient w.r.t. scaling factor: $\frac{\partial \overline{L}}{\partial \alpha}$

    • Compute gradient w.r.t. activations: $\frac{\partial \overline{L}}{\partial \boldsymbol{a}} = \frac{\partial \overline{L}}{\partial \tilde{\boldsymbol{a}}} h'(\boldsymbol{a})$

    • Compute gradient w.r.t. weights: $\frac{\partial \overline{L}}{\partial \boldsymbol{w}}$

      **Momentum-based update rule**                                 *// Novelty 1*

      **for** each layer in NN **do**

    • Compute first moment of gradient:

$$\boldsymbol{v}^{k+1} = \beta \boldsymbol{v}^k + (1 - \beta) \nabla_{\boldsymbol{w}} \overline{L}(\boldsymbol{w}^k, \alpha_k)$$

      **Kernel Rule**                                             *// Novelty 2, 3*

    • Compute statistics of gradient magnitudes:

$$l = \text{mean}(|v_i^{k+1}|), \quad \sigma = \text{std}(|v_i^{k+1}|)$$

    • Build update set:

$$\mathbb{S} = \left\{ i \mid w_i^k v_i^{k+1} > 0, \, \lambda_k \cdot \left| |v_i^{k+1}| - l \right| > \sigma \right\}$$

      **for** $i$ in $\mathbb{S}$ **do** $w_i^{k+1} \leftarrow -w_i^k$ **end for**

    **end for**

    Update $\alpha_k$ using any optimizer with LR $\mu_k$

  **end while**

---

### 3.4 RATIONALE FOR KERNEL RULE

We can think of top-k — a naive idea for weight flip rule, i.e. flipping the $k$ weights with the largest gradient, with scheduling $k$ to decrease during training. However, it is easy to come up with an example of a function for which the top-k method will not lead to the global optimum, since small gradients can flip important weights. Another important issue to consider with the top-k method is the choice of initial and last $k$ and the scheduler.

To avoid the necessity of choosing $k$ and its schedulere, we decided to adopt Chebyshev's inequality to filter weights that have important gradients. In theoretical research, we will show that if weights indices are from a set $\mathbb{S}'$ similar to $\mathbb{S}$ (the update set of KBOP), with $\mathbb{S}'$ requiring the knowledge of Lipschitz constant, then the loss will decrease during training. But using $\mathbb{S}'$ in practice, when Lipschitz constant is unknown, is impossible, therefore we choose to estimate Lipschitz constant with summation of gradients. To ensure the consistency of gradient signals, which is important in BNNs (Helwegen et al., 2019a), we adopted the first moment of gradient in KBOP.

## 4 THEORETICAL STUDY OF KBOP

In section 3 we introduce the BNN Init procedure. The conducted analysis is based on the work of He et al. (2015). Specifically, we derived initial scaling factor for binary weights drawn from Bernoulli distribution. Then, in the following subsection we provide the convergence guarantee for KBOP. Related proofs can be found in the supplementary materials.

### 4.1 BNN INITIALIZATION

The following theorems guarantees numerical stability during forward and backward propagation and proposes BNN Init strategy.

**Theorem 1.** *To assure the non-changing magnitudes of input signals in forward propagation, i.e.* $\mathrm{Var}(y_l) = \mathrm{Var}(y_{l-1})$, *where $y_i$ is the output of $i$-th layer, of a BNN with ReLU activation function it is sufficient to initialize binary weights $w_l \in \left\{ -\sqrt{2/n_l}, \sqrt{2/n_l} \right\}$ of a layer $l$ with $n_l$ weights using Bernoulli distribution with $p = 1/2$.*

**Theorem 2.** *To assure the non-changing magnitudes of input signals in backward propagation, i.e.* $\mathrm{Var}\left( \frac{\partial L}{\partial x_l} \right) = \mathrm{Var}\left( \frac{\partial L}{\partial x_{l+1}} \right)$, *where $L$ is the loss function and $x_i$ is the input of $i$-th layer, of a BNN with ReLU activation function it is sufficient to initialize binary weights $w_l \in \left\{ -\sqrt{2/\hat{n}_l}, \sqrt{2/\hat{n}_l} \right\}, \hat{n}_l = k_l^2 d_l$ of a layer $l$ using Bernoulli distribution with $p = 1/2$.*

### 4.2 CONVERGENCE ANALYSIS OF KBOP

In this section we provide the convergence guarantee of KBOP for functions with Lipschitz continuous gradient. We define the convergence of algorithm in $T$ steps as the condition when weight vector $\boldsymbol{w}^{k+1}$ obtained in $(k+1)$-th iteration equals $\boldsymbol{w}^T$ in $k$-th iteration. First, we consider the case $\beta = 0, \alpha_k = 1 \ \forall k \in \mathbb{N} \cup \{0\}$. Then we also consider the case of an arbitrary $\alpha_k$.

**Theorem 3.** *Let $L : \mathbb{R}^d \to \mathbb{R}$ be a differentiable function with Lipschitz continuous gradient $\nabla f$. Then there exists a monotonic sequence of KBOP LRs $\{\lambda_k\}$ such that KBOP algorithm converges in finite number of steps.*

Let us consider a differentiable function $L : \mathbb{R}^k \times [-1, 1]^d \to \mathbb{R}$ of $(\alpha, \boldsymbol{w})$, such that there exist constants $C > 0, \delta > 0$ that satisfy the following condition: $\forall \alpha^* \in \{\alpha : \|\alpha\| > C\} \ \forall \boldsymbol{w} \in [-1, 1]^d :$ $\left\langle \alpha^*, \frac{\partial L}{\partial \alpha}(\alpha^*, \boldsymbol{w}) \right\rangle \geq \delta \left\| \frac{\partial L}{\partial \alpha}(\alpha^*, \boldsymbol{w}) \right\| \geq 0$. Also, assume that

$$M := \sup_{\|\alpha^*\| \leq C, \boldsymbol{w}^* \in \{-1, 1\}^d} \left\| \frac{\partial L}{\partial \alpha}(\alpha^*, \boldsymbol{w}^*) \right\| < +\infty.$$

Let $\alpha_t$ denote the value of $\alpha$ after the gradient descent step $t$, and let $\mu_t$ be the learning rate at step $t$.

**Lemma 1.** *Let $\exists \mu > 0 : \forall t : \mu_t < \mu$ and if $\|\alpha_t\| > C$ then*

$$\mu_t < 2\delta \left[ \max_{\boldsymbol{w} \in \{-1,1\}^d} \left\| \frac{\partial L}{\partial \alpha}(\alpha_t, \boldsymbol{w}) \right\| \right]^{-1}$$

*Let $\|\alpha_0\| < C$, then there exists a constant $E := C + M\mu \geq C > 0$ such that $\forall t : \|\alpha_t\| < E$.*

As a consequence of the Lemma 1, we define the compact set $\Pi = \{\|\alpha\| \leq E\} \times [-1, 1]^d$, which will serve as the domain in the upcoming theorem.

**Theorem 4.** *Let $\nabla L$ be a Lipschitz continuous on $\Pi$. Then there exists a monotonic sequence of KBOP LRs $\{\lambda_k\}$, and there exists constant $\mu^*$, such that if $\mu_k < \mu^*$ for all $k$, then KBOP algorithm for $\boldsymbol{w}$ with gradient descent for $\alpha$ generates a convergent sequence $L_k = L(\alpha_k, \boldsymbol{w}_k)$. Furthermore, the following statements are true:*

1. *$\forall i > j : (\alpha_i, \boldsymbol{w}_i) \neq (\alpha_j, \boldsymbol{w}_j) \implies L_i < L_j$.*

2. *$\lim_{k \to \infty} L_k \in L(\Pi)$.*

3. *$\lim_{k \to \infty} L_k = \inf L_k$.*

## 5 EXPERIMENTS, ANALYSIS AND DISCUSSIONS

### 5.1 SETTING DESCRIPTION

We evaluate KBOP using computer vision and NLP benchmarks on a single Tesla A100 GPU, training EDSR for image super-resolution, ResNet-18 for image recognition, and GPT-2 for language modeling. The rationale for choosing GPT-2 can be found in Appendix G. For super-resolution, models were trained on DIV2K and tested on Set5 and Set14, with EDSR variants adapted from prior binarization studies. GPT-2 was modified with binary convolutions and evaluated on IMDB sentiment classification and WikiText-2 language modeling. Baseline methods (IR-Net, LAB, BNN, BBCU, ReActNet) were applied without altering their original architectures, except where necessary for convolutional adaptation in NLP tasks. Training used $L_1$ loss for super-resolution, AdamW optimization with linear scheduling for NLP, and hyperparameters chosen or adjusted per baseline, with two-stage training for ReActNet and high-precision settings for unstable IR-Net runs. All the experiments were conducted 3 times to ensure statistical consistency. This setup highlights KBOP's versatility across domains while ensuring fair benchmarking against existing BNN approaches. The detailed description of used settings is given in Appendix C. All hyperparameters are described in Appendix D and Appendix E.

### 5.2 QUALITY METRICS OF BNN TRAINING

For quantifying image quality, we employed two standard metrics – Peak Signal-to-Noise Ratio (PSNR) and Structural Similarity Index Measure (SSIM). These metrics were calculated exclusively on the luminance (y) channel, as this component is most closely aligned with human visual perception. We evaluated language models using Accuracy for classification tasks and Perplexity for text generation tasks. To enable a more refined and comprehensive comparison of the models under investigation, we introduced three custom-designed metrics that are tailored to capture aspects of model performance not fully addressed by conventional metrics. A detailed description of the custom quality metrics ($\mu_q$ – FP Difference, $\mu_m$ – Relative Memory Usage, $\mu_p$ – Convergence Plateau Comparison) is provided in Appendix F. All the metrics were averaged across 3 runs.

### 5.3 COMPARISON WITH SoTA BNN TRAINING METHODS

KBOP achieves lower quality difference, demonstrating a mean $\mu_q$ improvement of 30.82%: 35.96% for EDSR, 45.04% for GPT-2, 11.47% for ResNet18. The best improvement is achieved by EDSR (IR-Net) 81.15%.

KBOP sustains high memory efficiency with an average $\mu_m$ of 73.95%: 83.44% for EDSR and 64.46% for GPT-2 ensuring consistent compression across architectures. The highest efficiency is

Table 3: ResNet18 task results. Superscripts denote the modification of base network topology according to a baseline method: [1]IR-Net, [2]BBCU, [3]ReActNet.

| | Imagenet | | CIFAR-10 |
|---|---|---|---|
| **Method** | **Top-1** | **Top-5** | **Top-1** |
| | **Acc.(%)** | **Acc.(%)** | **Acc.(%)** |
| FP | 69.6 | 89.2 | 92.64 |
| IR-Net (Qin et al., 2020) | 58.1 | 80.0 | 91.49 |
| KBOP[1] | **58.3** | **80.1** | **91.55** |
| LAB (Falkena et al., 2023) | 64.2 | 85.0 | 84.1 |
| KBOP[2] | **64.5** | **85.3** | **84.5** |
| ReActNet (Liu et al., 2020) | 65.5 | 86.1 | 92.31 |
| KBOP[3] | **65.8** | **86.5** | **92.5** |

Table 4: Image super-resolution task results. EDSR performance on Set5, Set14 datasets. Superscripts denote the modification of base network topology according to a baseline method: [1]IR-Net, [2]BBCU, [3]ReActNet.

| Topology | Method | Set5 | | Set14 | |
|---|---|---|---|---|---|
| | | **PSNR** | **SSIM** | **PSNR** | **SSIM** |
| | FP | 31.421 | 0.895 | 27.819 | 0.789 |
| | IR-Net | 21.761 | 0.588 | 21.104 | 0.519 |
| | KBOP[1] | **29.271** | **0.845** | **26.414** | **0.746** |
| EDSR | LAB | 28.738 | 0.833 | 26.05 | 0.737 |
| (x4 scale) | KBOP[2] | **29.263** | **0.846** | **26.444** | **0.749** |
| | BBCU | 30.906 | 0.886 | 27.527 | 0.781 |
| | KBOP[2] | **31.098** | **0.891** | **27.604** | **0.783** |
| | ReActNet | 29.302 | 0.84 | **26.477** | 0.741 |
| | KBOP[3] | **29.339** | **0.846** | 26.444 | **0.746** |

Table 5: LLM task results. GPT-2 performance on IMDB and WIKITEXT. Superscripts denote the modification of base network topology according to a baseline method: [1]IR-Net, [2]LAB, [3]ReActNet.

| Topology | Method | IMDB | WIKITEXT |
|---|---|---|---|
| | | **Accuracy** | **Perplexity** |
| | FP | 0.886 | 169.928 |
| | IR-Net | 0.757 | 915.623 |
| | KBOP[1] | **0.877** | **302.565** |
| GPT-2 | LAB | **0.858** | 380.318 |
| | KBOP[2] | 0.528 | **372.505** |
| | ReActNet | 0.869 | **275.87** |
| | KBOP[3] | **0.877** | 303.225 |

observed across the EDSR cases, balancing quality and memory consumption. In contrast, GPT-2 cases have lower values of $\mu_m$, indicating greater memory consumption.

KBOP ensures a faster convergence, demonstrating a mean $\mu_p$ improvement of 17.25%: 9.83% for EDSR, 24.67% for GPT-2. The best stability is achieved by GPT-2 (ReActNet), showing strong optimization behavior in large-scale text modeling. However, EDSR (IR-Net) shows weaker plateau behavior, indicating sensitivity to the scaling factor behavior in its training dynamics.

The distinguishing features of KBOP may be seen on figures 1-2 that were obtained on the EDSR (BBCU) setting with LinearLR applied for the scaling factor. Figures present EDSR last layer (№33) parameters changing during training.

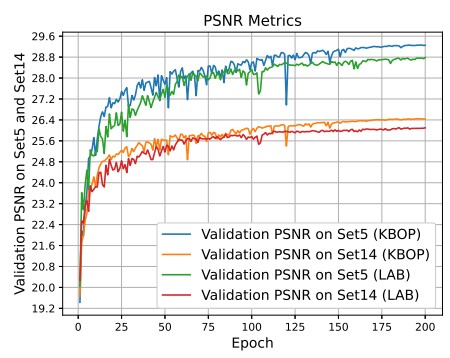

(a) PSNR metric dynamics on validation compared with the baseline, EDSR (LAB).

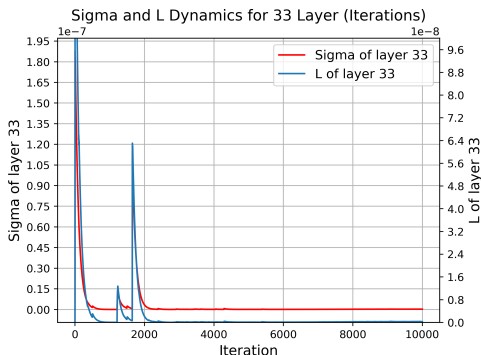

(b) $l$ (2) and $\sigma$ (3) for layer №33, EDSR (BBCU).

Figure 1: PSNR dynamics and the values of core statistics

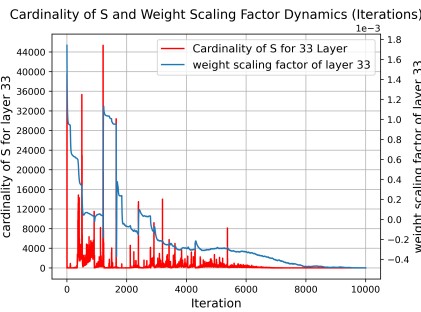

Figure 2: Cardinality of $\mathbb{S}$ (1) and weight scaling factor value for layer №33, EDSR (BBCU).

Figure 1 (a) shows PSNR change during the training. It is evident that in the last quarter of training (after 150 epoch) method achieves its plateau. Figure 1 (b) depicts change in the mean of weights absolute values (2) for layer № 33, indicating convergence of optimization process and change in the standard deviation of weights (3) for layer №33, indicating stabilization on the final stage of training process. Figure 2 shows the cardinality dynamics of $\mathbb{S}$ (1) for layer №33. It is clear that as the training progresses, weights begin to flip their signs less frequently. The same figure displays weight dynamics of scaling factor for layer №33, it is evident to be unstable during the first half of the training process and stable (i.e. has lower local variance) during the second half.

## 5.4 ABLATION STUDY

Table 6: KBOP ablation, ResNet18 on CIFAR-10 dataset.

| Case | Acc. (%) |
| --- | --- |
| FP | 0.939 |
| STE | 0.844 |
| KBOP | **0.857** |
| KBOP (without BNN Init) | 0.838 |
| KBOP (without KBOP LR & BNN Init) | 0.801 |
| KBOP (without KBOP LR) | 0.809 |

It can be seen (Table 6) that the proposed novelties (see Algorithm 2) lead to the increase in accuracy. Also it is evident that the aforementioned elements of the algorithm structure interact in a synergistic manner (which means that excluding any of them leads to a worse performance).

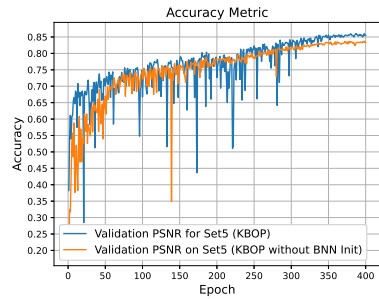

Figure 3: Accuracy metric values for KBOP with and without BNN Init, ResNet18 on CIFAR-10 dataset.

## 6 CONCLUSION

We proposed KBOP, a latent-free method of BNN training, which, without architectural modifications, improves training stability and quality in image SR, text classification and text generation tasks, and provided a theoretical convergence analysis. We also proposed a theoretically grounded BNN-specific initialization (BNN Init). Experimental results demonstrate consistent improvement in standard quality metrics. KBOP demonstrates lower average quality difference across the architectures, being 30.82% closer to FP than the SoTA BNN training baselines. For SR tasks, average PSNR improved by 2.157 and SSIM by 0.0825 over the baselines. For language modeling, accuracy increased by up to 4% on IMDB, and perplexity decreased by over 210 on WikiText.

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

# A    APPENDIX: BNN INIT DERIVATION

**Forward propagation.**    For a convolutional layer, a response is

$$\boldsymbol{y}_l = \boldsymbol{W}_l \boldsymbol{x}_l + \boldsymbol{b}_l. \tag{4}$$

Here, $\boldsymbol{x}$ is a $k^2 c$ dimensional vector that represents co-located $k \times k$ pixels in $c$ input channels. $k$ is the spatial filter size of the layer. With $n = k^2 c$ denoting the number of connections of a response, $\boldsymbol{W}$ is a $d \times n$ matrix with elements in $\{-\alpha, \alpha\}$, where $d$ is the number of filters and each row of $\boldsymbol{W}$ represents the weights of a filter. $\boldsymbol{b}$ is a vector of biases, and $\boldsymbol{y}$ is the response at a pixel of the output map. We use $l$ to index a layer. We obtain $\boldsymbol{x}_l = f(\boldsymbol{y}_{l-1})$, where $f$ is the activation, and $c_l = d_{l-1}$. We let the initialized elements in $\boldsymbol{W}_l$ be mutually independent and share the same distribution.

## A.1    PROOF OF THEOREM 1

*Proof.* Since the elements of $\boldsymbol{W}_l$ are binary and from $\{-\alpha_l, \alpha_l\}$, then their distribution is basically a Bernoulli distribution, so denote $p(w = \alpha_l)$, $(w = -\alpha_l)$ with $p_l$ and $q_l = 1 - p_l$ correspondingly. This leads to:

$$\mathbb{E}w = 2\alpha_l \left( p - \frac{1}{2} \right), Var(w) = 4\alpha^2 p - 4\alpha_l^2 p^2 - \alpha_l^2 + \alpha_l.$$

We assume that the elements in $\boldsymbol{x}_l$ are also mutually independent and share the same distribution, and $\boldsymbol{x}_l$ and $W_l$ are independent of each other. Then from (4) we have:

$$Var(y_l) = n_l Var(w_l x_l),$$

where now $y_l, x_l$, and $w_l$ represent the random variables of each element in $\boldsymbol{y}_l, W_l$, and $\boldsymbol{x}_l$ respectively. We let $w_l$ have zero mean, which is equivalent to $p_l = \frac{1}{2}$. Then the variance of the product of independent variables gives us:

$$Var(y_l) = n_l \left( \mathbb{E}\left[ w_l^2 \right] \mathbb{E}\left[ x_l^2 \right] - \mathbb{E}\left[ w_l \right]^2 \mathbb{E}\left[ x_l \right]^2 \right)$$

$$= n_l \alpha_l^2 \mathbb{E}\left[ x_l^2 \right]. \tag{5}$$

If we let $w_{l-1}$ have a symmetric distribution around zero and $b_{l-1} = 0$, then $y_{l-1}$ has zero mean and has a symmetric distribution around zero. Using $\boldsymbol{x}_l = f(\boldsymbol{y}_{l-1})$, we have:

$$\mathbb{E}\left[ x_l^2 \right] = \frac{1}{2} Var(y_{l-1}), \tag{6}$$

when f is ReLU. Putting (6) into (5), we obtain:

$$Var(y_l) = \frac{1}{2} n_l \alpha_l^2 Var(y_{l-1}).$$

With $L$ layers put together, we have:

$$Var(y_L) = Var(y_1) \prod_{l=2}^{L} \frac{1}{2} n_l \alpha_l^2.$$

This product is the key to the initialization design. A proper initialization method should avoid reducing or magnifying the magnitudes of input signals exponentially. So we expect the above product to take a proper scalar (e.g., 1). A sufficient condition is:

$$\frac{1}{2} n_l \alpha_l^2 = 1, \forall l = \overline{1; L}. \tag{7}$$

This leads to a zero-mean Bernoulli distribution $p(w_l = \alpha_l) = p(w_l = -\alpha_l) = \frac{1}{2}$, where $\alpha_l = \sqrt{\frac{2}{n_l}}$. We also initialize $\boldsymbol{b} = \boldsymbol{0}$. $\qquad \square$

## A.2 PROOF OF THEOREM 2

*Proof.* For backpropagation, the gradient of a convolution layer is computed by:

$$\Delta x_l = \hat{\boldsymbol{W}}_l \Delta y_l. \tag{8}$$

Here we use $\Delta x$ and $\Delta y$ to denote gradients $\left(\frac{\partial L}{\partial x}, \frac{\partial L}{\partial y}\right)$ for simplicity. $\Delta y$ represents $k \times k$ pixels in $d$ channels, and is reshaped into a $k^2 d$ vector. We denote $\hat{n} = k^2 d$. Note that $\hat{n} \neq n = k^2 c$. $\hat{\boldsymbol{W}}$ is a $c \times \hat{n}$ matrix where the filters are rearranged in the way of backpropagation. Note that $\boldsymbol{W}$ and $\hat{\boldsymbol{W}}$ can be reshaped from each other. $\Delta x$ is a vector of dimension $c$ representing the gradient at a pixel of this layer. As above, we assume that $w_l$ and $\Delta y_l$ are independent of each other, then $\Delta x_l$ has zero mean for all $l$, when $w_l$ is initialized by a symmetric distribution around zero ($p_l = \frac{1}{2}$).

In backpropagation we also have $\Delta y_l = f'(y_l)\Delta x_{l+1}$. For the ReLU case, $f'(y_l)$ is zero or one, and their probabilities are equal. We assume that $f'(y_l)$ and $\Delta x_{l+1}$ are independent of each other. Thus we have $\mathbb{E}\left[\Delta y_l\right] = \frac{1}{2}\mathbb{E}\left[\Delta x_{l+1}\right] = 0$, and also $\mathbb{E}\left[\Delta y_l^2\right] = Var\left(\Delta y_l\right) = \frac{1}{2}Var\left(\Delta x_{l+1}\right)$. Then we compute the variance of gradients in (8):

$$Var\left(\Delta x_l\right) = \hat{n}_l \alpha_l^2 Var\left(\Delta y_l\right) = \frac{1}{2}\hat{n}_l \alpha_l^2 Var\left(\Delta x_{l+1}\right).$$

With $L$ layers put together, we have:

$$Var\left(\Delta x_1\right) = Var\left(\Delta x_{L+1}\right)\prod_{l=2}^{L}\frac{1}{2}\hat{n}_l \alpha_l^2.$$

We consider a sufficient condition that the gradient is not exponentially large/small:

$$\frac{1}{2}\hat{n}_l \alpha_l^2 = 1, \forall l = \overline{1; L}. \tag{9}$$

The only difference between (9) and (7) is that $\hat{n}_l = k_l^2 d_l$, while $n_l = k_l^2 c_l = k_l^2 d_{l-1}$. Equation (9) leads to a zero-mean Bernoulli distribution $p(w_l = \alpha_l) = p(w_l = -\alpha_l) = \frac{1}{2}$, where $\alpha_l = \sqrt{\frac{2}{\hat{n}_l}}$. $\quad\square$

# B APPENDIX: KBOP CONVERGENCE PROOF

## B.1 PROOF OF THEOREM 3

*Proof.* Let $H$ be the Lipschitz constant for $\nabla L$. According to KBOP, weight $w_i^k$ is updated if $\frac{\partial L}{\partial w_i}\left(\boldsymbol{w}^k\right)w_i^k > 0$ and

$$\left|\frac{\partial L}{\partial w_i}\left(\boldsymbol{w}^k\right)\right| \leq l - \frac{\sigma}{\lambda}, \quad \text{or} \quad \left|\frac{\partial L}{\partial w_i}\left(\boldsymbol{w}^k\right)\right| \geq l + \frac{\sigma}{\lambda}.$$

Now we obtain the value of $\lambda$ that guarantees non-increase of $L$. For this purpose, we compute the mean of gradients at the current iteration

$$l = \frac{1}{d}\sum_{n=1}^{d}\left|\frac{\partial L}{\partial w_i}\left(\boldsymbol{w}^k\right)\right|.$$

Since $\forall k \in \mathbb{N}\ \boldsymbol{w}^k \in \{-1, 1\}^d$ and $\{-1, 1\}^d$ is finite, there exists an upper bound for the values of $\frac{\partial L}{\partial w_i}(\boldsymbol{w}^k)$. Denote this upper bound by $G$. Therefore, the following inequalities hold

$$|H - l| \leq H + l \leq H + G =: \varepsilon.$$

We show that if

$$\lambda < \min\left(\frac{\sigma}{\varepsilon}, \frac{\sigma}{H + l}\right),$$

then

$$\mathbb{S} = \left\{ i : \frac{\partial L}{\partial w_i} \left( \boldsymbol{w}^k \right) w_i^k > 0, \ \left| \left| \frac{\partial L}{\partial w_i} \left( \boldsymbol{w}^k \right) \right| - l \right| > \frac{\sigma}{\lambda} \right\}$$

$$\subset \left\{ i : \frac{\partial L}{\partial w_i} \left( \boldsymbol{w}^k \right) w_i^k > 0, \ \left| \frac{\partial L}{\partial w_i} \left( \boldsymbol{w}^k \right) \right| > H \right\} =: \mathbb{S}'.$$

For $\mathbb{S} \subset \mathbb{S}'$ to be true it is sufficient that the following inequalities hold:

$$l - \frac{\sigma}{\lambda} < -H, \qquad l + \frac{\sigma}{\lambda} > H. \tag{10}$$

Then, (13) is equivalent to

$$\lambda < \frac{\sigma}{H - l}, \qquad \lambda < \frac{\sigma}{H + l},$$

where the first inequality is required only if $H > l$. Equivalently, one may write

$$\lambda < \frac{\sigma}{|H - l|}, \qquad \lambda < \frac{\sigma}{H + l}.$$

Finally, since $|H - l| \leq \varepsilon$, it follows that

$$\frac{\sigma}{|H - l|} \geq \frac{\sigma}{\varepsilon},$$

so a sufficient condition ensuring both inequalities is

$$\lambda < \min \left( \frac{\sigma}{\varepsilon}, \ \frac{\sigma}{H + l} \right). \tag{11}$$

By the Lipschitz continuity of $L$

$$L \left( \boldsymbol{w}^k + \Delta \right) - L \left( \boldsymbol{w}^k \right) \leq \sum_{n=1}^{d} \frac{\partial L}{\partial w_i} \left( \boldsymbol{w}^k \right) \Delta_i + \frac{H}{2} \Delta_i^2, \tag{12}$$

where $\Delta \in \{-1, 1\}^d$. The right-hand side of (12) is either negative or $0$ under the update $\Delta$ according to KBOP. Indeed, if $w_i^k = -1$ and is updated, then, putting $\Delta_i^k = 2$ (the case $w_i^k = 1, \Delta_i^k = -2$ can be considered in the same manner), we get

$$\frac{\partial L}{\partial w_i} \left( \boldsymbol{w}^k \right) \Delta_i + \frac{H}{2} \Delta_i^2 = 2 \left( \frac{\partial L}{\partial w_i} \left( \boldsymbol{w}^k \right) + H \right) < 0,$$

since $\frac{\partial L}{\partial w_i} \left( \boldsymbol{w}^k \right) < l - \frac{\sigma}{\lambda} < -H$. Thus, we found the value of $\lambda$ that guarantees non-increase of $L$. To get a monotonic sequence of KBOP's LRs, we can set $\lambda_k = \min\{\lambda_{k-1}, \lambda_k^*\}$, where $\lambda_k^*$ satisfies the inequality (11). This completes the proof, since the feasible set of considered problem is finite. $\square$

## B.2 PROOF OF LEMMA 1

*Proof.* Let $E = C + M\mu$. If $M = 0$, there's nothing to prove, because if $M = 0$ then $\left\| \frac{\partial L}{\partial \alpha} \right\| = 0$ for $\|\alpha_t\| < C = E$. Let $M > 0$, $E > C$. We prove the lemma by induction. The base case $t = 0$ does not require verification, as it is one of the assumptions of the lemma.

Inductive step $t \to t + 1$:

1. Let $\|\alpha_t\| \leq C$ then

$$\|\alpha_{t+1}\| = \|\alpha_{t+1} - \alpha_t + \alpha_t\| \leq \|\alpha_t\| + \|\alpha_{t+1} - \alpha_t\| \leq C + \mu_k \| \frac{\partial L}{\partial \alpha} \| < C + M\mu = E.$$

2. Let $C < \|\alpha_t\| < E$.

$$\|\alpha_{t+1}\|^2 = \|\alpha_t\|^2 - 2\mu_t \left\langle \alpha_t, \frac{\partial L}{\partial \alpha} \right\rangle + \mu_t^2 \| \frac{\partial L}{\partial \alpha} \|^2 \leq \|\alpha_t\|^2 < E^2.$$

$\square$

As a consequence of the lemma, we define the compact set $\Pi = \{\|\alpha\| \leq E\} \times [-1, 1]^d$, which will serve as the domain in the upcoming theorem.

### B.3   PROOF OF THEOREM 4

*Proof.* Let $H$ be the Lipschitz constant for $\nabla L$.

Since $\nabla L$ is Lipschitz on $\Pi$, it follows that $\nabla L$ is continuous on $\Pi$, therefore exist $\tilde{M} = \max_\Pi \left\| \frac{\partial L}{\partial \alpha} \right\|$. Let $\mu_k < \min\left\{ \mu, 2\delta/\tilde{M}, 2/H \right\} =: \mu^*$. It is easy to see that $\mu_k$ satisfies the assumptions of Lemma 1, therefore $(\alpha^k, \boldsymbol{w}^k) \in \Pi$ for all $k$.

According to KBOP, weight $w_i^k$ is updated if $\frac{\partial L}{\partial w_i}\left(\alpha^k, \boldsymbol{w}^k\right) w_i^k > 0$ and

$$\left| \frac{\partial L}{\partial w_i}\left(\alpha^k, \boldsymbol{w}^k\right) \right| \le l - \frac{\sigma}{\lambda}, \quad \text{or} \quad \left| \frac{\partial L}{\partial w_i}\left(\alpha^k, \boldsymbol{w}^k\right) \right| \ge l + \frac{\sigma}{\lambda}.$$

Now we obtain the value of $\lambda$ that guarantees non-increase of $L$. For this purpose, we compute the mean of gradients at the current iteration

$$l = \frac{1}{d} \sum_{n=1}^{d} \left| \frac{\partial L}{\partial w_i}\left(\alpha^k, \boldsymbol{w}^k\right) \right|.$$

Since $(\alpha^k, \boldsymbol{w}^k) \in \Pi$, there exists an upper bound for the values of $\frac{\partial L}{\partial w_i}(\alpha^k, \boldsymbol{w}^k)$ by Weierstrass theorem. Denote this upper bound by $G$. Therefore, the following inequalities hold

$$|H - l| \le H + l \le H + G =: \varepsilon.$$

We show that if

$$\lambda < \min\left( \frac{\sigma}{\varepsilon}, \frac{\sigma}{H + l} \right),$$

then

$$\mathbb{S} = \left\{ i : \frac{\partial L}{\partial w_i}(\alpha^k, \boldsymbol{w}^k) w_i^k > 0, \left| \left| \frac{\partial L}{\partial w_i}(\alpha^k, \boldsymbol{w}^k) \right| - l \right| > \frac{\sigma}{\lambda} \right\}$$
$$\subset \left\{ i : \frac{\partial L}{\partial w_i}(\alpha^k, \boldsymbol{w}^k) w_i^k > 0, \left| \frac{\partial L}{\partial w_i}(\alpha^k, \boldsymbol{w}^k) \right| > H \right\} =: \mathbb{S}'.$$

For $\mathbb{S} \subset \mathbb{S}'$ to be true it is sufficient that the following inequalities hold:

$$l - \frac{\sigma}{\lambda} < -H, \qquad l + \frac{\sigma}{\lambda} > H. \tag{13}$$

Then, (13) is equivalent to

$$\lambda < \frac{\sigma}{H - l}, \qquad \lambda < \frac{\sigma}{H + l},$$

where the first inequality is required only if $H > l$. Equivalently, one may write

$$\lambda < \frac{\sigma}{|H - l|}, \qquad \lambda < \frac{\sigma}{H + l}.$$

Finally, since $|H - l| \le \varepsilon$, it follows that

$$\frac{\sigma}{|H - l|} \ge \frac{\sigma}{\varepsilon},$$

so a sufficient condition ensuring both inequalities is

$$\lambda < \min\left( \frac{\sigma}{\varepsilon}, \frac{\sigma}{H + l} \right). \tag{14}$$

By the Lipschitz continuity of $L$

$$L\left(\alpha^k + \Delta_\alpha, \boldsymbol{w}^k + \Delta\right) - L\left(\alpha^k, \boldsymbol{w}^k\right) \le \sum_{i=1}^{d} \left( \frac{\partial L}{\partial w_i}\left(\boldsymbol{w}^k\right) \Delta_i + \frac{H}{2}\Delta_i^2 \right) + \\ + \sum_i \left( \frac{\partial L}{\partial w_i}\left(\alpha^k, \boldsymbol{w}^k\right) \Delta_{\alpha,i} + \frac{H}{2}\Delta_{\alpha,i}^2 \right), \tag{15}$$

where $\Delta \in \{-2, 0, 2\}^d$, $\Delta_\alpha = -\mu_k \frac{\partial L}{\partial \alpha}$.

The first sum in right-hand side of (15) is either negative or 0 under the update $\Delta$ according to KBOP. Indeed, if $w_i^k = -1$ and is updated, then, putting $\Delta_i^k = 2$ (the case $w_i^k = 1, \Delta_i^k = -2$ can be considered in the same manner), we get

$$\frac{\partial L}{\partial w_i}\left(\alpha^k, \boldsymbol{w}^k\right)\Delta_i + \frac{H}{2}\Delta_i^2 = 2\left(\frac{\partial L}{\partial w_i}\left(\alpha^k, \boldsymbol{w}^k\right) + H\right) < 0,$$

since $\frac{\partial L}{\partial w_i}\left(\boldsymbol{w}^k\right) < l - \frac{\sigma}{\lambda} < -H$.

The second sum in right-hand side of (15) is either negative or 0, because $0 < \mu_k < \dfrac{2}{H}$. Indeed,

$$\frac{\partial L}{\partial \alpha_i}\left(\alpha^k, \boldsymbol{w}^k\right)\Delta_{\alpha,i} + \frac{H}{2}\Delta_{\alpha,i}^2 = \frac{\partial L}{\partial \alpha_i}^2 \mu_k\left(-1 + \mu_k\frac{H}{2}\right) \leq 0,$$

Moreover, if $\frac{\partial L}{\partial \alpha} \neq 0$, then this equation is negative.

Thus, we found the value of $\lambda$ that guarantees:

$$L\left(\alpha^k + \Delta_\alpha, \boldsymbol{w}^k + \Delta\right) - L\left(\alpha^k, \boldsymbol{w}^k\right) \leq 0.$$

Consider the sequence $f_k = L(\alpha_k, \boldsymbol{w}_k)$. Since $L$ is continuous on $\Pi$ and $(\alpha_k, \boldsymbol{w}_k) \in \Pi$, we have $f_k \in L(\Pi)$. That is, $f_k$ is a sequence of points in the compact set $L(\Pi)$, and therefore it has a limit $L^* = \inf f_k$ and $L^* \in L(\Pi)$, since this is a non-increasing sequence of points in a compact set.

To get a monotonic sequence of KBOP's LRs, we can set $\lambda_k = \min\{\lambda_{k-1}, \lambda_k^*\}$, where $\lambda_k^*$ satisfies the inequality (14).

$\square$

# C  APPENDIX: SETTING DESCRIPTION

In our research, one Tesla A100 GPU was used to train EDSR (Lim et al., 2017a), GPT-2 (Radford et al., 2019a) and ResNet-18 (He et al., 2016).

**Datasets and neural network architectures description.**   We evaluate KBOP using image super-resolution, image recognition and natural language processing benchmarks. Super-resolution models are trained on the DIV2K (Agustsson & Timofte, 2017) dataset, which comprises 800 high-quality 2K images for training, along with 100 validation and 100 test images. For testing, we used two popular benchmarks: Set5 (Bevilacqua et al., 2012), consisting of five widely recognized images, and Set14 (Zeyde et al., 2012), which includes 14 images showcasing a diverse range of scenes and textures. We conducted experiments on the standard EDSR (Lim et al., 2017a) and its baseline variants with architectural changes from recent binarization studies, applying KBOP as a training method while preserving each baseline's architecture. The EDSR architecture, in contrast, removes batch normalization, employs 32 residual blocks with ReLU activations, and contains roughly 43 million parameters. To evaluate the generalizability of our binarization approach, we extended our study to the NLP domain by modifying the GPT-2 model. The base GPT-2 architecture includes 12 transformer layers, 12 attention heads, and a hidden size of 768, amounting to approximately 117 million parameters. We replaced standard convolutions in the architecture with binary convolutions to examine their effect on performance. To ensure a comprehensive evaluation of GPT-2, we conducted tests on various tasks, including binary classification (IMDB) (Maas et al., 2011) and text generation (WikiText-2) (Merity et al., 2016). The IMDB dataset comprises 50,000 movie reviews labeled for sentiment. WikiText-2 includes 2 million tokens extracted from Wikipedia articles and is used for language modeling. This multi-domain evaluation highlights KBOP versatility across computer vision and NLP tasks and offers broader benchmarking than prior BNN studies.

**Benchmarking settings.**   To apply our method, we incorporated the architectural modifications in the neural network according to the baseline models while adjusting the training process and convolutional approximations. In the super-resolution task, we treated the network architecture the same way as the baseline methods. IR-Net, LAB, and BNN (Courbariaux et al., 2016) do not introduce any architectural changes, so no modifications were required. BBCU employs RSign, RPReLU,

additional residual connections, and bilinear approximation in the final layer, while ReActNet utilizes RSign, RPReLU, and additional residual connections. Since LAB focuses solely on activation binarization, we applied the ReActNet method for weight binarization, following the approach used in the LAB-BNN model described in (Falkena et al., 2023). For the GPT-2 language model, we adopted only the RSign mechanism from BBCU and ReActNet. Additionally, we replaced Conv1d convolutions with adapted versions specific to each method, effectively transforming Conv2d into Conv1d for language models while preserving the core principles of each approach.

**Experimental hyperparameters.** In image super-resolution task we focus on 4x image quality enhancement, aiming to increase the resolution of input images by a factor of four. Each mini-batch consists of 16 images, randomly cropped from the training set. In the BBCU, IR-Net, LAB, ReActNet, and STE, whereas in all other cases, a 192x192 image size was employed. For training, we utilize the $L_1$ loss as our primary metric to ensure accurate pixel-wise reconstruction and promote sharper image details. Table 7 and Table 8 outline the hyperparameters, where "Base NN" indicates the backbone model used and baseline method with its corresponding architectural adjustments. IR-Net, LAB, and ReActNet methods primarily focused on the image recognition task; therefore, for the super-resolution and NLP tasks, the hyperparameters were chosen by us to ensure efficient training. Additionally, the ReactNet model was trained in two stages as described by Liu et al. (2020). The scheduler was reinitialized at the beginning of each stage. In the case of IR-Net EDSR, 64-bit precision was used due to training instability.

For natural language processing experiments, we employ the well-established GPT-2 architecture as the basic model, with modifications implemented through the substitution of its binary convolution layers. A default linear LR schedule is applied across all experimental runs. Additionally, we consistently utilize the AdamW optimizer, configured with betas of (0.9, 0.999) and an epsilon value of 1e-8. Table 8 summarizes the hyperparameters used in our experiments, where the term "Base NN" is defined as in the preceding table.

## D  APPENDIX: BASELINES HYPERPARAMETERS

Table 7: Hyperparameters for EDSR

| Base NN | Optimizer (Adam) | Scheduler |
|---|---|---|
| EDSR (IR-Net, STE) | `betas=(0.9, 0.999), lr=1e-4` | `StepLR( step_size=50, gamma=0.5)` |
| EDSR (Re-Act) | `params=filter( lambda p: p.requires_grad), lr=1e-4` | `StepLR( step_size=150, gamma=0.5)` |
| EDSR (LAB) | `betas=(0.9, 0.999), lr=1e-4` | `StepLR( step_size=55, gamma=0.5)` |
| EDSR (BBCU) | `betas=(0.9, 0.999), lr=2e-4` | `Cosine AnnealingLR( T_max=200, eta_min=1e-7)` |

`lr` - LR, `betas` - coefficients used for computing running averages of gradient and its square, `step_size` - period of LR decay, `gamma` - multiplicative factor of LR decay, `T_max` - maximum number of iterations, `eta_min` - minimum LR. Number of epochs is set to `200`, `lr_range` and `hr_range` are both set to `(0;1)`.

Table 8: Hyperparameters for GPT-2

| Hyperpar. | IMDB | WIKITEXT |
|---|---|---|
| Base NN | GPT-2 (IR-Net/ STE), GPT-2 (ReActNet/ BBCU/ LAB) | GPT-2 (IR-Net/ STE), GPT-2 (ReActNet/ BBCU/LAB) |
| Optimizer (AdamW) | lr=5e-4, lr=5e-5 | lr=5e-4, lr=5e-5 |
| Scheduler | LinearLR | LinearLR |
| Epochs | 10 | 50 |
| Batch Size | 8 | 7 |

Table 9: Hyperparameters for CLF model.

| Dataset | Baseline NN | Optimizer | Scheduler | Epochs | Batch |
|---|---|---|---|---|---|
| CIFAR-10 | ResNet-18 (IR-Net/STE) | Adam betas=(0.9, 0.99), lr=1e-2 | CosineAnnealingLR (eta_min=1e-7) | 400 | 128 |
| Imagenet | ResNet-18 (IR-Net/STE) | SGD momentum=0.9, weight decay=1e-5, lr=0.2 | StepLR (step_size=18750, gamma=0.1) | 1 | 64 |
| CIFAR-10 | ResNet-18 (LAB) | Adam betas=(0.9, 0.99), lr=1e-3 | CosineAnnealingLR (eta_min=1e-7) | 200 | 128 |
| Imagenet | ResNet-18 (LAB) | Adam betas=(0.9, 0.999), lr=2.5e-3 | CosineAnnealingLR (eta_min=0) | 300 | 256 |
| CIFAR-10 | ResNet-18 (ReActNet) | Adam betas=(0.9, 0.999), lr=5e-4 | LambdaLR( lambda: 1-epoch/epochs) | 120 | 512 |
| Imagenet | ResNet-18 (ReActNet) | Adam betas=(0.9, 0.999), lr=5e-4 | LambdaLR( lambda: 1-epoch/epochs) | 120 | 512 |

## E  APPENDIX: KBOP HYPERPARAMETERS

Table 10: KBOP hyperparameters for EDSR

| Base NN | Scheduler | Scailing factor lr scheduler | KBOP Scheduler | KBOP params |
|---|---|---|---|---|
| EDSR (IR-Net) | StepLR( step_size=50, gamma=0.5) | LinearLR initial_lr=1e-4 final_lr=0 | LambdaLR(lr_lambda = lambda step: 1 if step<150 else math.sqrt(0.05)/0.3) | lr = 0.3 momentum=0.999 |
| EDSR (ReAct) | StepLR( step_size=150, gamma=0.5) | LinearLR initial_lr=1e-4 final_lr=0 | LambdaLR(lr_lambda = lambda step: 1 if step<150 else 0.7453) | lr=0.3 momentum=0.999 |
| EDSR (LAB) | StepLR( step_size=55, gamma=0.5) | LinearLR initial_lr=2e-4 final_lr=0 | Cosine AnnealingLR( eta_min=0.883) | lr =0.936 momentum=0.999 |
| EDSR (BBCU) | Cosine AnnealingLR( T_max=200, eta_min=1e-7) | Same as in the previous column. | Cosine AnnealingLR( eta_min=0.01) | lr = 0.8 momentum=0.99 |

Number of epochs is set to 200, lr_range and hr_range are both set to (0;1).

Table 11: KBOP hyperparameters for GPT WIKITEXT

| Base NN | Scheduler | Scailing factor lr scheduler | KBOP Scheduler | KBOP params |
|---|---|---|---|---|
| GPT (IR-Net, ReAct, LAB) | LinearLR | `LinearLR initial_lr=1e-4 final_lr = 0` | `LambdaLR(lr_lambda = lambda step: 1 if step<5810 else math.sqrt(0.05)/0.3)` | `lr = 0.3 momentum=0.999` |

Table 12: KBOP hyperparameters for GPT IMDB

| Base NN | Scheduler | Scailing factor lr scheduler | KBOP Scheduler | KBOP params |
|---|---|---|---|---|
| GPT (IR-Net, ReAct, LAB) | LinearLR | `LinearLR initial_lr=1e-4 final_lr = 0` | `LambdaLR(lr_lambda = lambda step: 1 if step<1563*7 else math.sqrt(0.05)/0.3)` | `lr = 0.3 momentum=0.999` |

Table 13: KBOP hyperparameters for ResNet18 CIFAR-10

| Base NN | KBOP Scheduler | KBOP params |
|---|---|---|
| ResNet18 (IR-Net, Re-Act, LAB) | `Cosine AnnealingLR( eta min=0.01)` | `lr = 0.1 momentum=0.99` |

Table 14: KBOP hyperparameters for ResNet18 ImageNet

| Base NN | KBOP Scheduler | KBOP params |
|---|---|---|
| ResNet18 (IR-Net, Re-Act, LAB) | `Cosine AnnealingLR( eta min=0.1)` | `lr = 0.3 momentum=0.99` |

## F  APPENDIX: QUALITY METRICS

For quantifying image quality, we employed two standard metrics – Peak Signal-to-Noise Ratio (PSNR) and Structural Similarity Index Measure (SSIM). These metrics were calculated exclusively on the luminance (y) channel, as this component is most closely aligned with human visual perception. We evaluated language models using a two-pronged approach. For classification tasks, Accuracy was used as the primary metric to assess model performance. In contrast, for text generation tasks, we adopted Perplexity as the evaluation metric. To enable a more refined comparison of the BNN training methods, we introduced three custom-designed metrics that are tailored to capture aspects of model performance not fully addressed by conventional metrics.

**Quality Difference.**   This metric is designed to reflect the superiority of performance value of one model over another in percentage terms. It shows how much the metric gap between the binary model being tested and the full precision model has been reduced relative to the baseline: $\mu_q = \frac{kbop\_result - baseline\_result}{fp\_result - baseline\_result} \times 100(\%)$, where $kbop\_result$ is the performance metric of KBOP, $baseline\_result$ is the performance metric of the baseline method. A positive value of $\mu_q$ indicates that the proposed approach improves performance compared to the baseline, while a negative value signifies a drop in performance.

**Relative Memory Usage.**   This metric of relative memory usage is the index of the size difference between the given binarized model and its full precision counterpart: $\mu_m = 0.9375 n_b/n_t \times 100(\%)$, where $n_b$ – total amount of binary weights, $n_t$ – total amount of binary and full precision weights. A higher percentage implies a more significant reduction in memory usage.

**Plateau Comparison.** This plateau comparison metric is designed to show the superiority of one plateau over another in percentage terms. Plateau is an epoch number at which the stopping criterion was triggered: $\mu_p = \frac{baseline\_plateau - kbop\_plateau}{baseline\_plateau} \times 100(\%)$, where $baseline\_plateau$ – epoch at which the metric of the baseline reaches its maximum value, $kbop\_plateau$ – epoch at which the metric of KBOP reaches its maximum value.

## G NLP EXPERIMENTS SETTING

For our NLP experiments, we employ the well-established GPT-2 architecture with modifications implemented through the substitution of its binary convolution layers. We chose GPT-2 as our primary benchmark for the following reasons:

- GPT-2 (and other models of similar scale) has become the de facto intermediate-scale baseline in recent quantization studies, serving as a common reference point for comparing methods and enabling meta-analysis across works (Chitsaz et al., 2024; Park et al., 2022; Frantar et al., 2023; Xu et al., 2024; Dettmers & Zettlemoyer, 2023). Based on this experience, we can assert the applicability of our proposal for LLM using only this model. We also note that there are no works on the binarization of LLM of this size (up to a billion), which means that we are the first to propose such a benchmark, which can open up access to this area for most of the scientific community.
- GPT-2 weights are openly available and require only modest compute resources (e.g., a single high-memory GPU). This ensures that our experiments can be easily reproduced by the community without the need for large-scale clusters (Radford et al., 2019b).

## H APPENDIX: FINE-TUNING KBOP PARAMETERS

For fine-tuning the KBOP parameters (initial and final KBOP lr, KBOP momentum), we used the TPE implementation from the Optuna library with default settings and n_trials = 10; each trial was run for one third of an epoch, and the comparison was based on the metrics obtained at that point.

KBOP momentum was tuned in the range 0.9–0.999, the initial learning rate in the range 0.1–1, and the final learning rate in the range $\sqrt{0.01}$–$\sqrt{0.08}$.

