# OpenReview forum: "Kernel Binary Optimizer (KBOP): Latent-free Optimizer for Binary Neural Networks"
_ICLR.cc/2026/Conference — ICLR 2026 Conference Withdrawn Submission_

### Official Review · Reviewer_jS4k · 2025-10-29

**Soundness:** 2
**Presentation:** 2
**Contribution:** 2
**Rating:** 4
**Confidence:** 4

**Summary:**

This paper proposes a novel training method for Binary Neural Networks (BNNs), named Kernel Binary Optimizer (KBOP), aiming to enhance training stability and performance. The authors claim that KBOP achieves improvements in tasks such as image super-resolution, text classification, and text generation without architectural modifications. They also provide theoretical convergence analysis. While the experiments cover diverse architectures and datasets, demonstrating performance gains, the paper has notable issues in experimental design, theoretical rigor, and result interpretation that require further clarification.

**Strengths:**

(1) Proposes KBOP, a new BNN training method that allegedly improves stability and quality without modifying the architecture.
(2) Designs a BNN-specific initialization method (BNN Init) combined with a tailored learning rate scheduling strategy.
(3) Provides theoretical convergence guarantees and validates KBOP through experiments on multiple tasks and datasets.

**Weaknesses:**

(1) Insufficient experimental comparison: The claim that KBOP is "36.83% closer to FP (full-precision) than SoTA BNN baselines" lacks a clear definition of the calculation method. Full comparison results with all baseline methods are missing, weakening the credibility of this assertion.
(2) Overly simplified theoretical analysis: Although convergence theorems (Theorem 3 and 4) are presented, the paper does not explain why KBOP’s learning rate scheduling ensures convergence. The proofs lack critical technical details, making it hard to verify the theoretical validity.
(3) Questionable result comparability: Table 6 reports a 0.857 accuracy on CIFAR-10, but no comparison with state-of-the-art methods is provided. The statistical significance of the performance gap over baselines is also unaddressed.
(4) Missing implementation details: While "KBOP with BNN Init" outperforms variants, the paper fails to explain how BNN Init is implemented or why it was chosen, hindering reproducibility.
(5) Unexplained hyperparameter settings: Tables 10–14 list hyperparameter combinations but do not justify their selection or analyze their impact on performance, reducing the interpretability of results.

**Questions:**

(1) How does KBOP specifically improve over existing BNN training methods (e.g., IR-Net, LAB, BBCU)? The paper claims "KBOP demonstrates lower average quality difference across architectures, being 36.83% closer to FP than SoTA baselines," but no concrete comparison data or calculation methodology is provided.
(2) The paper states that "KBOP (without BNN Init) achieves 0.838" compared to the full KBOP’s 0.857. Why is BNN Init critical? What are the specific implementation details of BNN Init, and why does it significantly boost performance?
(3) The paper claims KBOP "improves PSNR by an average of 2.157," but it does not specify which baseline methods this comparison refers to or provide per-dataset results. This undermines the credibility of the reported performance gain.

---

> ### Author Response · Authors · 2025-11-22
> **Rebuttal by Authors (part 1)**
>
> # **Response to Reviewer jS4k**
>
> We thank the reviewer for the careful reading of our manuscript and the constructive comments.
>
> ---
>
> ## **Weakness 1**
>
> > *“Insufficient experimental comparison: The claim that KBOP is "36.83% closer to FP (full-precision) than SoTA BNN baselines" lacks a clear definition of the calculation method...”*
>
> **Response:**
>
> Thank you for your feedback! The closeness-to-fp metric $\mu_q$ is defined in Appendix F and is computed as (kbop_result − baseline_result) / (fp_result − baseline_result), where kbop_result is the performance metric of KBOP, baseline_result is the performance metric of the baseline method. Comparison data is presented in Tables 3, 4, 5.
>
> ---
>
> ## **Weakness 2**
>
> > *"Overly simplified theoretical analysis: Although convergence theorems (Theorem 3 and 4) are presented, the paper does not explain why KBOP’s learning rate scheduling ensures convergence. The proofs lack critical technical details, making it hard to verify the theoretical validity."*
>
> **Response:**
>
> We are sorry that the paper has a broken equations references, which certainly makes it hard to understand the proofs. We have fixed this issue in the revised paper.
>
> As for the learning rate schedule, the lines 706–734 and 792–816 (in the first version) (753–777 and 831–857in the revised version) contain the proof that one can schedule $\lambda$ such that it satisfies the inequality which leads to the negativity of loss delta (lines 743-745 in the first version, 784–786 in the revised version). To get a monotonically decreasing sequence of KBOP’s LRs, we can set $\lambda_k$ = $\min ( \lambda_{k-1}, \lambda_k^* )$ where $\lambda_k^*$ satisfies inequalities in lines 733 and 816 in the first version (775, 856 in the revised version). We forgot to add the last step to the proof, but we have added it during the revision. We hope that this clarification will make it easier to understand.
>
> ---
>
> ## **Weakness 3**
> > *"Questionable result comparability: Table 6 reports a 0.857 accuracy on CIFAR-10, but no comparison with state-of-the-art methods is provided. The statistical significance of the performance gap over baselines is also unaddressed."*
>
> **Response:**
>
> Thank you for the comment! Table 6 contains an ablation study, carried out under a simple image classification setting. Сomparison with SOTA on image classification is presented in Table 3. The statistical significance of the performance gap follows from the fact that Tables 3-5 contain the mean performance metric values for 3 consecutive runs. However, this was not clarified in the paper, which was addressed during the revision.
>
> ---
>
> ## **Weakness 4**
>
> > *"Missing implementation details: While "KBOP with BNN Init" outperforms variants, the paper fails to explain how BNN Init is implemented or why it was chosen, hindering reproducibility."*
>
> **Response:**
>
> The motivation and theoretical description of BNN Init are presented in Sections 3.2 and 4.1. Implementation: for each layer, binary weights were sampled from a Bernoulli distribution with $p = 0.5$, and the scaling factor was set to $\sqrt{\frac{2}{n}}$. We have included the corresponding pseudocode in Section 3.2.
>
> ---
>
> ## **Weakness 5**
>
> > *"Unexplained hyperparameter settings: Tables 10–14 list hyperparameter combinations but do not justify their selection or analyze their impact on performance, reducing the interpretability of results."*
>
> **Response:**
>
> Thank you for your comment! The TPE Bayesian optimization algorithm was used for hyperparameter tuning (initial and final KBOP LR, KBOP momentum). We are sorry that this was not clarified in the paper. We have added Section H to the appendix, which provides a more detailed description of the hyperparameter fine-tuning.

---

> ### Author Response · Authors · 2025-11-22
> **Rebuttal by Authors (part 2)**
>
> # **Q/A**
>
> ## **Q1. How does KBOP specifically improve over existing BNN training methods (e.g., IR-Net, LAB, BBCU)?..**
>
> The closeness-to-fp metric $\mu_q$ is defined in Appendix F and is computed as (kbop_result − baseline_result) / (fp_result − baseline_result), where kbop_result is the performance metric of KBOP, baseline_result is the performance metric of the baseline method. Comparison data is presented in Tables 3, 4, 5.
>
> ---
>
> ## **Q2. The paper states that "KBOP (without BNN Init) achieves 0.838" compared to the full KBOP’s 0.857. Why is BNN Init critical?..**
>
> In non-convex optimization, and especially in binary optimization, the choice of initial conditions is no less important than the optimization algorithm itself. BNN Init makes the outputs and gradients of homogeneous layers homogeneous and also improves numerical stability (the variances and means of the layer outputs match the variances and means of their inputs, as shown in Section 4.1), which is what leads to the improvement. Implementation: for each layer, binary weights were sampled from a Bernoulli distribution with $p = 0.5$, and the scaling factor was set to $\sqrt{\frac{2}{n}}$. We have included the corresponding pseudocode in Section 3.2.
>
> ---
>
> ## **Q3. The paper claims KBOP "improves PSNR by an average of 2.157," but it does not specify which baseline methods this comparison refers to or provide per-dataset results...**
>
> The mentioned claim refers to the data presented in Table 4. The final value shown in the paper is calculated as an average of all PSNR improvements across each baseline and dataset addressed in the table.

---

### Official Review · Reviewer_3F7Y · 2025-11-01

**Soundness:** 3
**Presentation:** 2
**Contribution:** 2
**Rating:** 4
**Confidence:** 5

**Summary:**

This paper proposes an optimizer for binary networks called KBOP which flips binary weights based on an adaptive and statistic based threshold derived from gradient distributions. KBOP also introduces a specialized initialization scheme for binary networks. KBOP is designed to enable stable and direct training of the value of binary networks without using latent real valued weights.

**Strengths:**

1. Clear motivation and problem statement
2. Novel idea advanced from existing works:
- Kernel rule: While binary optimizer uses a simple constant threshold, KBOP introduces the effect of dynamic and statistical threshold using the mean and standard deviation of the gradient magnitudes across a layer instead of a fixed number.
- BNN initialization: Inspired by the fact that binary weights are fixed at +1 or -1 with real valued scaling factor for each layer, KBOP derives optimal initial value for this scaling factor using Kaiming analysis which used to derive Kaiming initialization.

While the proposed methods are novel, I have a lot of concerns about the results.

**Weaknesses:**

I integrated this section to Questions section.

**Questions:**

1. The amount of quality improvement with KBOP is shown small. In some cases, it doesn’t show improvement.
2. In the comparison with Reactnet ResNet-18 Imagenet case, why don’t you compare it on MobileNet? How did you construct the block structure in this case? As you may know the base structure of Reactnet is designed from MobileNet. I think this comparison is important since Reactnet is sota.
2. Are you sure that the perplexity of wikitext on GPT-2 FP is 169.93?
3. As in Table 6, when KBOP isn’t conducted with BNN init, the result is inferior than STE. Does it mean that the kernel rule and other proposed dynamic scalings have no improvement?
4. In Table 6, what happens if BNN init is applied on STE?

Minor
- Typo of superscript ‘4’ in Table5.

---

> ### Author Response · Authors · 2025-11-22
> **Rebuttal by Authors (part 1)**
>
> # **Response to Reviewer 3F7Y**
>
> We sincerely thank the reviewer for their thoughtful comments and constructive feedback.
>
> ---
>
> # **Q/A**
>
> ## **Q1. The amount of quality improvement with KBOP is shown small. In some cases, it doesn’t show improvement.**
>
> We would like to clarify three crucial points that show the quality improvement with KBOP is strong, which is supported directly by the experimental results and comparison with quality improvement of other SoTA BNN algorithms.
>
> (1) KBOP consistently achieves state-of-the-art or best-in-class results across all three domains we evaluate.
> - Vision classification: KBOP achieves the best results among all BNN baselines on both Imagenet and CIFAR-10 (Table 4)
> - Super-resolution: KBOP outperforms baselines in nearly all PSNR/SSIM settings for EDSR.
> - NLP: KBOP outperforms baselines in nearly all cases for GPT-2 (Table 5).
>
> (2) A core goal of KBOP is not only to reach top accuracy, but to *substantially reduce the performance gap between binarized and full-precision networks.*
>
> Unlike prior methods that focus on a single architecture or setting, KBOP provides a unified training framework that works *without modifying* the underlying architecture and can be applied to any baseline.
> When measuring the *proximity interval* between SOTA and full-precision models, KBOP provides ≈30.82% interval coverage ($\mu_q$), meaning that it recovers *part* of the accuracy lost in binarization. This metric more accurately reflects the purpose of a general binarization framework.
>
> Below we provide the average interval coverage between SOTA and full-precision models for other SOTA algorithms to show that KBOP provides a considerable improvement to close the gap between BNNs and FP networks.
>
> **AdaBin**
>
> *ImageNet:*
>
> - ResNet-18: FP 69.6, AdaBin 63.1, baseline 61.0 (ReCU)
>   - $\mu_q$ = 100 * (63.1 - 61.0) / (69.6 - 61.0) = 24.41%
> - ResNet-34: FP 73.3, AdaBin 66.4, baseline 65.1 (ReCU)
>   - $\mu_q$ = 100 * (65.4 - 65.1) / (73.3 - 65.1) = 3.65%
> - AlexNet: FP 56.6, AdaBin 53.9, baseline 50.5 (SiBNN)
>   - $\mu_q$ = 100 * (53.9 - 50.5) / (56.6 - 50.5) = 55.73%
>
> *Cifar10:*
>
> - ResNet-18: FP 94.8, AdaBin 93.1, baseline 92.8 (ReCU)
>   - $\mu_q$ = 100 * (93.1 - 92.8) / (94.8 - 92.8) = 15%
> - ResNet-20: FP 91.7, AdaBin 88.2, baseline 87.8 (RBNN)
>   - $\mu_q$ = 100 * (88.2 - 87.8) / (92.7 - 87.8) = 8.16%
> - VGG-Small: FP 94.1, AdaBin 92.3, baseline 92.0 (SLB)
>   - $\mu_q$ = 100 * (92.3 - 92.0) / (94.1 - 92.0) = 14.28%
>
> **Average $\mu_q$ = 20.205%**
>
> ---
>
> **ReBNN**
>
> *ImageNet:*
>
> - ResNet-18 (one-stage training): FP 69.6, ReBNN 61.6, baseline 61.0 (ReCU)
>   - $\mu_q$ = 100 * (61.6 - 61.0) / (69.6 - 61.0) = 6.96%
> - ResNet-18 (two-stage training): FP 69.6, ReBNN 66.9, baseline 66.4 (ReCU)
>   - $\mu_q$ = 100 * (66.9 - 66.4) / (69.6 - 66.4) = 15.62%
> - ResNet-34 (one-stage training): FP 73.3, ReBNN 65.8, baseline 65.1 (ReCU)
>   - $\mu_q$ = 100 * (65.8 - 65.1) / (73.3 - 65.1) = 8.53%
> - ResNet-34 (two-stage training): FP 73.3, ReBNN 69.9, baseline 69.3 (ReActNet)
>   - $\mu_q$ = 100 * (69.9 - 69.3) / (73.3 - 69.3) = 15%
>
> **Average $\mu_q$ = 11.52%**
>
> ---
>
> **VISPA**
>
> *ImageNet:*
>
> - ResNet-18: FP 69.6, VISPA 62.1, baseline 61.6 (ReBNN)
>   - $\mu_q$ = 100 * (62.1 - 61.6) / (69.6 - 61.6) = 6.25%
> - AlexNet: FP 56.6, VISPA 51.1, baseline 47.9 (Quantization Networks)
>   - $\mu_q$ = 100 * (51.1 - 47.9) / (56.6 - 47.9) = 36.78%
>
> *Cifar10:*
>
> - ResNet-18: FP 94.8, VISPA 92.8, baseline 92.8 (ReCU, DIR-Net, RBNN + CMIM)
>   - $\mu_q$ = 100 * (92.8 - 92.8) / (94.8 - 92.8) = 0%
> - VGG-Small: FP 94.1, VISPA 92.7, baseline 92.6 (ReSTE)
>   - $\mu_q$ = 100 * (92.7 - 92.6) / (94.1 - 92.6) = 6.66%
>
> **Average $\mu_q$ = 12.42%**
>
> ---
>
> **LAB**
>
> *ImageNet:*
>
> - ResNet-18: FP 69.6, LAB-BNN 64.2, baseline 63.3 (QuickNet)
>   - $\mu_q$ = 100 * (64.2 - 63.3) / (69.6 - 63.3) = 14.28%
>
> **Average $\mu_q$ = 14.28%**
>
> ---
>
> **IR-Net**
>
> *ImageNet:*
>
> - ResNet-18: FP 69.6, IR-Net 66.5, baseline 64.3 (BWHN)
>   - $\mu_q$ = 100 * (66.5 - 64.3) / (69.6 - 64.3) = 41.50%
> - ResNet-34 (Bi-Real structure): FP 73.3, IR-Net 62.9, baseline 62.2 (Bi-Real Net)
>   - $\mu_q$ = 100 * (62.9 - 62.2) / (73.3 - 62.2) = 6.30%
>
> *Cifar10:*
>
> - ResNet-18: FP 93.0, IR-Net 91.5, baseline 90.5 (RAD)
>   - $\mu_q$ = 100 * (91.5 - 90.5) / (93.0 - 90.5) = 40%
> - ResNet-20: FP 91.7, IR-Net 85.4, baseline 84.1 (DSQ)
>   - $\mu_q$ = 100 * (85.4 - 84.1) / (92.7 - 84.1) = 15.11%
> - VGG-Small: FP 91.7, IR-Net 90.4, baseline 90.0 (RAD)
>   - $\mu_q$ = 100 * (90.4 - 90.0) / (91.7 - 90.0) = 23.52%
>
> **Average $\mu_q$ = 25.286%**

---

> ### Author Response · Authors · 2025-11-22
> **Rebuttal by Authors (part 2)**
>
> (3) KBOP uniquely improves multiple existing binarization approaches through plug-and-play integration.
>
> Since KBOP and BNN Init can be inserted into any existing binary network without architectural changes, KBOP consistently *boosts* IR-Net, LAB, BBCU, ReActNet, and STE. Thus, evaluating KBOP solely by absolute accuracy overlooks its primary contribution: a general, architecture-agnostic mechanism that enhances other binarization pipelines.
>
> Together, these points show that KBOP not only achieves strong accuracy, but also establishes a generalizable and integrative binarization strategy that narrows the FP–BNN gap more effectively than existing methods.
>
> ---
>
> ## **Q2. In the comparison with Reactnet ResNet-18 Imagenet case, why don’t you compare it on MobileNet?..**
>
> The basic ReActNet modification approach remains the same (Sign is replaced by ReActNet Sign, ReLU is replaced by RPReLU), the convolution blocks are replaced with modified ReActNet blocks (where 1 x 1 convolution is changed to 3 x 3 convolution), the structure of an individual identity block remains the same as in the ReActNet paper, with pairs of 3 x 3 depth-wise and 1 x 1 point-wise convolutions being replaced by a ReActNet block. This leads to a seamless transfer of the ReActNet approach to the ResNet18 topology.
>
> ---
>
> ## **Q3. Are you sure that the perplexity of wikitext on GPT-2 FP is 169.93?**
>
> Thank you for raising this concern. Yes, we confirm that the reported perplexity of 169.93 corresponds to our own GPT-2 (FP) model trained from scratch, not to the community OpenAI GPT-2 checkpoint. To clarify the discrepancy:
> - We trained GPT-2 from scratch on the WikiText-2 dataset, specifically the wikitext-2-raw-v1 split, which contains approximately 2 million tokens in total (≈18 MB of text).
> - Training was performed for 50 epochs (as reported in Table 8). This means that the model effectively “saw” roughly 100 million tokens during training.
> - In contrast, the original GPT-2 model published at https://huggingface.co/openai-community/gpt2 was trained on a much larger corpus of ~40 GB, which is roughly ≈2275× larger than the dataset used in our setting.
> - Naturally, a model trained on 2M tokens cannot reach the perplexity levels of a model trained on 40G of text. Our results reflect a *fair, controlled comparison of binarization methods under identical training conditions*, not an attempt to reproduce the original GPT-2 training scale.
>
> Our choice of WikiText-2 was driven by two constraints:
> - Hardware: All language-model experiments were conducted on a single NVIDIA A100 40GB.
> - Reproducibility: We intentionally used a small, publicly available dataset so that all baselines and the proposed KBOP method can be reproduced easily by the community without requiring large-scale computation.
>
> ---
>
> ## **Q4. As in Table 6, when KBOP isn’t conducted with BNN init, the result is inferior than STE. Does it mean that the kernel rule and other proposed dynamic scalings have no improvement?**
>
> The superiority of STE over KBOP without BNN Init is explained by the fact that STE (a latent-weight algorithm) was initialized using the Kaiming distribution. As for KBOP without BNN Init, an unsuitable initialization was used: weights were initialized in full precision with a uniform distribution (pytorch default initialization), then a scaling factor for the layer was obtained as the average absolute value of the weights in that layer, and after that the sign function was applied to the weights. In non-convex optimization, and particularly in binary optimization, the choice of initial conditions is no less important than the optimization algorithm itself. Nevertheless, other ablation cases demonstrate that with and without proper initialization, the proposed novelties provide an improvement.
>
> ---
>
> ## **Q5. In Table 6, what happens if BNN init is applied on STE?**
>
> STE is a latent-weight algorithm. In other words, it works with full precision weights during training, which is why it's better to use a pure Kaiming distribution (the foundation of BNN Init) rather than BNN Init itself, since the core idea of latent-weight algorithms is to replace the binary network with a full-precision network during training, and as a result, numerical stability of the forward and backward passes becomes critical for training such networks, just as it is for regular non-binary networks.
>
> On the other hand, initialization of the scaling factor is an important part of BNN Init, but in the case of latent-weight algorithms the scaling factor is not a separate learnable parameter of the network (it is computed from the latent weights), so formally it is not applicable to latent-weight algorithms at all. In our paper, Kaiming initialization was used for the STE.
>
> ## **Minor. Typo of superscript ‘4’ in Table5.**
>
> We appreciate your careful reading. We have corrected the typo in the revised paper.

---

### Official Review · Reviewer_xoDg · 2025-11-03

**Soundness:** 3
**Presentation:** 3
**Contribution:** 2
**Rating:** 4
**Confidence:** 2

**Summary:**

The paper proposes a novel method to train Binary Neural Networks (BNN) with binar weights and activations. It is called kernel binary optimizer (KBOP) and uses a heuristic based on the first and second order moments of the exponential moving average computed over the gradients to control which weights should flip sign during training. The proposed selection rule has two parameters. $\lambda$ can be interpreted as the learning rate and $\beta$ is the parameter of the exponential moving average over the weight gradients.

**Strengths:**

The paper gives a decent overview of the performance of recent and old methods to train BNNs in section 2.

**Weaknesses:**

Several aspects limit its clarity of the paper:

1) Key aspects of the algorithm that are required to understand and reproduce it are missing or scattered in the paper. For example, the criterion proposed to select which parameters flip sign is based on statistics of $v$, the exponential moving average of the gradient. However, this one is only defined in the pseudo code. It is not clear how the moments are calculated.

2) Important information about the practical behavior of the algorithm are missing. First, there is little information about how to chose the parameters $\lambda$ and $\beta$. Choices for the experiments are not documented or at least hidden to me. Second, the complexity of the algorithm is not discussed?

3) Inconsistent performance results are given in tables what diminishes my trust in the results (see questions)

**Questions:**

1) How is $l=E[|v_i|]$ estimated? Is it $l=\frac{1}{d} 1^Tv$? If yes, the followup question would be if it is computed across all parameters or for each layer separately, because I am a bit concerned about selection bias across layers.

2) What is the rationale behind the proposed selection rule, because the choice feels a bit arbitrary. I can think of many other selection methods, e.g. we could just flip the top-k weights receiving largest gradients and schedule k to decrease. Because the selection method is the core proposal, I miss some theory or experiments why it should be optimal.

3) In Table 1, the best SOTA method is ReActNet wit $69.4%$ top-1 accuracy. However, in Table 3, it only achieves $65.5%$. I guess this is a typo.

4) I looked into the ReActNet paper and actually they even report $71.4%$ with their ReActNet-C, which is much higher then your reported $65.8%$ with KBOP. I guess there is a reason you did not compare to ReActNet-C, but I would like to know to trust the results.

---

> ### Author Response · Authors · 2025-11-22
> **Rebuttal by Authors**
>
> # **Response to Reviewer xoDg**
> We are grateful to the reviewer for their time and expertise.
>
> ---
>
> ## **Weakness 1**
>
> > *"Key aspects of the algorithm that are required to understand and reproduce it are missing or scattered in the paper..."*
>
> **Response:**
> The gradient momentum is computed as $v_{k+1} = \beta v_k + (1-\beta) \nabla L$, that is, in the same way as, for example, in [1]. Thank you for the comment; we have described this in more detail in the Section 3.1 in new revision of the paper.
>
> **References:**
> - [1] Kingma, D. P., & Ba, J. (2014). Adam: A method for stochastic optimization.
>
> ---
>
> ## **Weakness 2**
>
> > *"Important information about the practical behavior of the algorithm are missing."*
>
> **Response:**
> We deeply appreciate your observations. The TPE Bayesian optimization algorithm was used for hyperparameter tuning (initial and final KBOP lr, KBOP momentum). We have added Section H to the appendix, which provides a more detailed description of the hyperparameter fine-tuning. Hyperparameters are presented in Appendix E.
>
> Computing the mean and variance from the gradient moments has an asymptotic complexity of O(n), where n is the number of weights. Updating the weights also has an asymptotic complexity of O(n), so the asymptotic complexity of a KBOP step is the same as that of SGD step for a given objective function.
>
> ---
>
> ## **Weakness 3**
>
> > *"Inconsistent performance results are given in tables what diminishes my trust in the results (see questions)."*
>
> **Response:**
> Thank you for pointing out the inconsistency in our original script. We sincerely apologize, the current version of the paper contains wrong numerical results, which is due to the typo made in the process of formatting the final table. The correct values are presented in the following table. We once again thank you for noticing this mistake, the correct data was added in the main paper (Tables 1 and 3) during the revision.
>
> ---
>
> # **Q/A**
>
> ## **Q1. How is $l = E[v_i]$ estimated? Is it $l = \frac{1}{d}1^T v$?..**
>
> Yes, it is $\frac{1}{d} 1^{T}v$. $l$ is computed for each layer separately, this is written in line 241 in Pseudocode 2. We are very grateful for your feedback!
>
> ---
>
> ## **Q2. What is the rationale behind the proposed selection rule, because the choice feels a bit arbitrary...**
>
> We deeply appreciate the reviewer’s observation. It is worth mentioning that we have tested the naive top-k flip method based on gradient magnitudes during the research, but it did not show satisfactory performance. One can come up with an example of a function for which the top-k method will not lead to the global optimum, since small gradients can flip important weights. Another important issue to consider with the top-k method is the choice of $k$.
>
> To avoid the necessity to choose $k$, we decided to adopt Chebyshev's inequality to filter weights that have important gradients. In theoretical research, specifically in lines 735-748 (in the first version of the paper) and lines 777–790 (in the revised version), we showed that if weights indices are from $\mathbb{S}’$, then the loss delta is negative. But using $\mathbb{S}’$ in practice, when Lipschitz constant is unknown, is impossible, therefore we choose to estimate Lipschitz constant with summation of gradients. To ensure the consistency of gradient signals, which is important in BNNs [1], we adopted the first moment of gradient in KBOP.
>
> We understand that the rationale might be unclear when reading the paper. We have included a thorough description for the rationale in Section 3.4.
>
> **References:**
> - [1] Helwegen, K., Widdicombe, J., Geiger, L., Liu, Z., Cheng, K. T., & Nusselder, R. (2019). Latent weights do not exist: Rethinking binarized neural network optimization
> ---
>
> ## **Q3. In Table 1, the best SOTA method is ReActNet with $69.4$ top-1 accuracy...**
>
> We appreciate your comment! ReActNet was tested only on MobileNet architecture in the original paper. These results were added in Table 1 (ResNet18 on ImageNet dataset), which is clearly a typo. We have removed the ReActNet row from Table 1. The results that are presented in Table 3 were obtained by us by applying ReActNet architectural modifications (RSign, RPReLU) and two-stage training to ResNet18.
>
> ---
>
> ## **Q4. I looked into the ReActNet paper and actually they even report $71.4$ with their ReActNet-C, which is much higher then your reported  with KBOP...**
>
> Thank you for your observation! ReActNet-C replaces the 1-bit 1×1 convolution with real-valued 1×1 convolution in the downsampling layers. We considered fully binarized weight and activations in our benchmarks, hence only ReActNet-A was appropriate for the comparison.

---

> > ### Comment · Reviewer_xoDg · 2025-11-25
> >
> > Thank you for answering my questions and for making the required corrections. About
> >
> > **Q2:** Since you also thought about and tried the top-k selection method, I think it would make sense to include this discussion in the paper (if only in the appendix).
> >
> > **Q4:** I understand that the comparison with ReActNet-C would not be fare, since it uses more powerful high-precision 1x1 convolutions. Maybe a footnote would be appropriate to add this information to the manuscript.
> >
> > Your clarifications helped me to build trust in the experiments. However, improvements upon SOTA seem to be minor. Hence, I tend to keep my current score.

---

> > > ### Author Response · Authors · 2025-11-25
> > > **Official Comment by Authors (part 1)**
> > >
> > > Thank you for the response! We have already added the discussion about top-k selection method in Section 3.4 of the revised paper. About the ReActNet-C, we have inserted a specific description of ReActNet-A in 920 line in the paragraph about benchmarking settings.
> > >
> > > We would like to clarify three crucial points that show the quality improvement with KBOP is strong, which is supported directly by the experimental results and comparison with quality improvement of other SoTA BNN algorithms.
> > >
> > > (1) KBOP consistently achieves state-of-the-art or best-in-class results across all three domains we evaluate.
> > > - Vision classification: KBOP achieves the best results among all BNN baselines on both Imagenet and CIFAR-10 (Table 4)
> > > - Super-resolution: KBOP outperforms baselines in nearly all PSNR/SSIM settings for EDSR.
> > > - NLP: KBOP outperforms baselines in nearly all cases for GPT-2 (Table 5).
> > >
> > > (2) A core goal of KBOP is not only to reach top accuracy, but to *substantially reduce the performance gap between binarized and full-precision networks.*
> > >
> > > Unlike prior methods that focus on a single architecture or setting, KBOP provides a unified training framework that works *without modifying* the underlying architecture and can be applied to any baseline.
> > > When measuring the *proximity interval* between SOTA and full-precision models, KBOP provides ≈30.82% interval coverage ($\mu_q$), meaning that it recovers *part* of the accuracy lost in binarization. This metric more accurately reflects the purpose of a general binarization framework.

---

> > > ### Author Response · Authors · 2025-11-25
> > > **Official Comment by Authors (part 2)**
> > >
> > > Below we provide the average interval coverage between SOTA and full-precision models for other SOTA algorithms to show that *KBOP provides a considerable improvement to close the gap between BNNs and FP networks.*
> > >
> > > **AdaBin**
> > >
> > > *ImageNet:*
> > >
> > > - ResNet-18: FP 69.6, AdaBin 63.1, baseline 61.0 (ReCU)
> > >   - $\mu_q$ = 100 * (63.1 - 61.0) / (69.6 - 61.0) = 24.41%
> > > - ResNet-34: FP 73.3, AdaBin 66.4, baseline 65.1 (ReCU)
> > >   - $\mu_q$ = 100 * (65.4 - 65.1) / (73.3 - 65.1) = 3.65%
> > > - AlexNet: FP 56.6, AdaBin 53.9, baseline 50.5 (SiBNN)
> > >   - $\mu_q$ = 100 * (53.9 - 50.5) / (56.6 - 50.5) = 55.73%
> > >
> > > *Cifar10:*
> > >
> > > - ResNet-18: FP 94.8, AdaBin 93.1, baseline 92.8 (ReCU)
> > >   - $\mu_q$ = 100 * (93.1 - 92.8) / (94.8 - 92.8) = 15%
> > > - ResNet-20: FP 91.7, AdaBin 88.2, baseline 87.8 (RBNN)
> > >   - $\mu_q$ = 100 * (88.2 - 87.8) / (92.7 - 87.8) = 8.16%
> > > - VGG-Small: FP 94.1, AdaBin 92.3, baseline 92.0 (SLB)
> > >   - $\mu_q$ = 100 * (92.3 - 92.0) / (94.1 - 92.0) = 14.28%
> > >
> > > **Average $\mu_q$ = 20.205%**
> > >
> > > ---
> > >
> > > **ReBNN**
> > >
> > > *ImageNet:*
> > >
> > > - ResNet-18 (one-stage training): FP 69.6, ReBNN 61.6, baseline 61.0 (ReCU)
> > >   - $\mu_q$ = 100 * (61.6 - 61.0) / (69.6 - 61.0) = 6.96%
> > > - ResNet-18 (two-stage training): FP 69.6, ReBNN 66.9, baseline 66.4 (ReCU)
> > >   - $\mu_q$ = 100 * (66.9 - 66.4) / (69.6 - 66.4) = 15.62%
> > > - ResNet-34 (one-stage training): FP 73.3, ReBNN 65.8, baseline 65.1 (ReCU)
> > >   - $\mu_q$ = 100 * (65.8 - 65.1) / (73.3 - 65.1) = 8.53%
> > > - ResNet-34 (two-stage training): FP 73.3, ReBNN 69.9, baseline 69.3 (ReActNet)
> > >   - $\mu_q$ = 100 * (69.9 - 69.3) / (73.3 - 69.3) = 15%
> > >
> > > **Average $\mu_q$ = 11.52%**
> > >
> > > ---
> > >
> > > **VISPA**
> > >
> > > *ImageNet:*
> > >
> > > - ResNet-18: FP 69.6, VISPA 62.1, baseline 61.6 (ReBNN)
> > >   - $\mu_q$ = 100 * (62.1 - 61.6) / (69.6 - 61.6) = 6.25%
> > > - AlexNet: FP 56.6, VISPA 51.1, baseline 47.9 (Quantization Networks)
> > >   - $\mu_q$ = 100 * (51.1 - 47.9) / (56.6 - 47.9) = 36.78%
> > >
> > > *Cifar10:*
> > >
> > > - ResNet-18: FP 94.8, VISPA 92.8, baseline 92.8 (ReCU, DIR-Net, RBNN + CMIM)
> > >   - $\mu_q$ = 100 * (92.8 - 92.8) / (94.8 - 92.8) = 0%
> > > - VGG-Small: FP 94.1, VISPA 92.7, baseline 92.6 (ReSTE)
> > >   - $\mu_q$ = 100 * (92.7 - 92.6) / (94.1 - 92.6) = 6.66%
> > >
> > > **Average $\mu_q$ = 12.42%**
> > >
> > > ---
> > >
> > > **LAB**
> > >
> > > *ImageNet:*
> > >
> > > - ResNet-18: FP 69.6, LAB-BNN 64.2, baseline 63.3 (QuickNet)
> > >   - $\mu_q$ = 100 * (64.2 - 63.3) / (69.6 - 63.3) = 14.28%
> > >
> > > **Average $\mu_q$ = 14.28%**
> > >
> > > ---
> > >
> > > **IR-Net**
> > >
> > > *ImageNet:*
> > >
> > > - ResNet-18: FP 69.6, IR-Net 66.5, baseline 64.3 (BWHN)
> > >   - $\mu_q$ = 100 * (66.5 - 64.3) / (69.6 - 64.3) = 41.50%
> > > - ResNet-34 (Bi-Real structure): FP 73.3, IR-Net 62.9, baseline 62.2 (Bi-Real Net)
> > >   - $\mu_q$ = 100 * (62.9 - 62.2) / (73.3 - 62.2) = 6.30%
> > >
> > > *Cifar10:*
> > >
> > > - ResNet-18: FP 93.0, IR-Net 91.5, baseline 90.5 (RAD)
> > >   - $\mu_q$ = 100 * (91.5 - 90.5) / (93.0 - 90.5) = 40%
> > > - ResNet-20: FP 91.7, IR-Net 85.4, baseline 84.1 (DSQ)
> > >   - $\mu_q$ = 100 * (85.4 - 84.1) / (92.7 - 84.1) = 15.11%
> > > - VGG-Small: FP 91.7, IR-Net 90.4, baseline 90.0 (RAD)
> > >   - $\mu_q$ = 100 * (90.4 - 90.0) / (91.7 - 90.0) = 23.52%
> > >
> > > **Average $\mu_q$ = 25.286%**
> > >
> > > (3) KBOP uniquely improves multiple existing binarization approaches through plug-and-play integration.
> > >
> > > Since KBOP and BNN Init can be inserted into any existing binary network without architectural changes, KBOP consistently *boosts* IR-Net, LAB, BBCU, ReActNet, and STE. Thus, evaluating KBOP solely by absolute accuracy overlooks its primary contribution: a general, architecture-agnostic mechanism that enhances other binarization pipelines.
> > >
> > > Together, these points show that KBOP not only achieves strong accuracy, but also establishes a generalizable and integrative binarization strategy that narrows the FP–BNN gap more effectively than existing methods.

---

> > > > ### Comment · Reviewer_xoDg · 2025-11-26
> > > >
> > > > Thank you for your work and the additional clarification. I will contemplate about your response and get back to you.

---

### Official Review · Reviewer_xoRM · 2025-11-04

**Soundness:** 3
**Presentation:** 3
**Contribution:** 2
**Rating:** 6
**Confidence:** 3

**Summary:**

This paper introduces a latent-free optimization (called Kernel Binary Optimizer) for training binary neural networks. To address the mismatch of the non-contiguity nature of BNNs and the contiguous latency weights used by gradient based optimizers, three major changes are made to the weight updates rule
Rather than frequently flipping the weight signs, they proposed a sign flipping rule based on the accumulated binary gradients. For the binary weight updates, it uses a separate learning rate
The learnable layer-wise scaling factor is using a separate LR to reduce quantization error.
They proposed a theoretically derived initialization (BNN Init) ensuring variance stability for binary weights.

**Strengths:**

1. Good demonstration of effectiveness on multiple applications.
2. Simple kernel rule is for weight update

**Weaknesses:**

The accuracy drop for imagenet (65.8% vs. 69.6%) seems large.

**Questions:**

How are different LRs (initial learning rate) tuned for different tasks?
Accuracy comparison of SoTA BNN training methods for image classification task and super resolution tasks is shown. How about  language tasks?

---

> ### Author Response · Authors · 2025-11-22
> **Rebuttal by Authors**
>
> # **Response to Reviewer xoRM**
>
> We thank the reviewer for the careful reading of our manuscript and the constructive comments.
>
> ---
>
> ## **Weakness 1**
>
> > *“The accuracy drop for imagenet (65.8% vs. 69.6%) seems large.”*
>
> **Response:**
> Full binarization of weights and activations of neural networks necessarily leads to performance degradation, caused by the limited representative ability of binary weights and activations in comparison with floating-point weights and activations [1]. It is a general problem of BNNs. Even with such an accuracy drop, KBOP shows the best performance amongst SoTA BNN algorithms.
>
> References:
> - [1] Yuan, C., & Agaian, S. S. (2023). A comprehensive review of binary neural network
>
> ---
>
> # **Q/A**
>
> ## **Q1. How are different LRs (initial learning rate) tuned for different tasks?**
>
> The TPE Bayesian optimization algorithm was used for hyperparameter tuning (initial and final KBOP lr, KBOP momentum), initial learning rate for the full-precision parameters was tuned for the baselines and then carried over to KBOP without any changes. We have added Section H to the appendix, which provides a more detailed description of the hyperparameter fine-tuning.
>
> ---
>
> ## **Q2. Accuracy comparison of SoTA BNN training methods for image classification task and super resolution tasks is shown. How about language tasks?**
> The language modeling results are already included in Table 2.
>
> We evaluate KBOP on two standard language tasks using GPT-2, a transformer-based architecture: IMDB (text classification), WikiText-2 (language modeling, perplexity). For transformer architecture we choose GPT‑2 as our primary benchmark for the following reasons:
> - GPT‑2 (and other models of similar scale) has become the de facto intermediate‑scale baseline in recent quantization studies, serving as a common reference point for comparing methods and enabling meta‑analysis across works [1, 2, 3, 4, 5]. Based on this experience, we can assert the applicability of our proposal for LLM using only this model. We also note that there are no works on the binarization o fLLM of this size (up to a billion), which means that we are the first to propose such a benchmark, which can open up access to this area for most of the scientific community.
> - GPT‑2 weights are openly available and require only modest compute resources (e.g., a single high‑memory GPU). This ensures that our experiments can be easily reproduced by the community without the need for large‑scale clusters [6]. We are actively working on extending KBOP to larger LLMs and will report these results in a follow‑up.
>
>
> Across both tasks (language modeling and classification), KBOP achieves the best performance in 4/6 cases among all BNN baselines.
> These results already demonstrate that KBOP generalizes beyond vision tasks and performs strongly on language tasks, including transformer-based models.
> To make this clearer for readers, we have revised the manuscript so that the choice of GPT-2 as baseline is highlighted more explicitly in the Appendix G (reference is placed in Section 5.1).
>
> **References:**
> - [1] Chitsaz K. et al. Exploring Quantization for Efficient Pre‑Training of Transformer Language Models.
> - [2] Park M. et al. Quadapter: Adapter for GPT‑2 Quantization.
> - [3] Frantar E. et al. GPTQ: Accurate Post‑Training Quantization for Generative Pre‑Trained Transformers.
> - [4] Xu Z. et al. Scaling Laws for Post‑Training Quantized Large Language Models.
> - [5] Dettmers T., Zettlemoyer L. The case for 4-bit precision: k-bit inference scaling laws
> - [6] Radford et al. Language Models are Unsupervised Multitask Learners (GPT‑2 Release)

---

### Note · Authors · 2026-03-12

**Comment:**

We are issuing a withdrawal because the work was published without our knowledge. To avoid any misunderstandings, we would like to clarify: the rejection was recieved on January 26th.

**Withdrawal Confirmation:**

I have read and agree with the venue's withdrawal policy on behalf of myself and my co-authors.

---

### Meta-Review · Area_Chair_w1MG · 2026-01-01

**Summary:**

The paper presents a procedure for training binary neural networks, with a few components such as a kernel rule and an initialization rule. The authors also supplemented their approach with a theoretical analysis

Reviewers share a few major concerns
1. The clarify of the paper, including technical details
2. The performance improvement is incremendal.

I tried to read the paper and found the paper has significant issues:

1. Key concepts are not explained. For example, I am very confused about what a kernel is, since "kernel" is such a common terminology that has many meanings, but they paper did not provide a clear definition.
2. The methodology section (namely, Secs 3.1 and 3.2) mainly presents a procedure of their method, without providing rationales. So I am not getting an intuitive picture of the benefits.
3. The paper is full of typos and erros:
* Lines 147-149: w in R^d, a in R^p, z=wa. The dimensions seem to mismatch. Inner-product of w and a is not typically written as wa.
* Line 157: backward propagation algorithm -> the backward propagation algorithm
* Line 162: Q_a represent -> Q_a represents
* Line 174: left quotation mark -> right quotation mark
* Line 176: does not works -> does not work
* Line 177:  Other latent-free optimizers, such as Bop (Helwegen et al., 2019b) and Bop2Order (Suarez-Ramirez et al., 2021) uses -> use

While I highly commend the authors for not using AI tools in their writing (at least not extensively), the paper needs to be thoroughly proofread before it's ready to be considered for publication.

**Reviewer Concerns:**

Reviewers share a few major concerns
1. The clarify of the paper, including technical details
2. The performance improvement is incremendal.

**Reviewer Scores:**

I tried to read the paper and found the paper has significant issues:

1. Key concepts are not explained. For example, I am very confused about what a kernel is, since "kernel" is such a common terminology that has many meanings, but they paper did not provide a clear definition.
2. The methodology section (namely, Secs 3.1 and 3.2) mainly presents a procedure of their method, without providing rationales. So I am not getting an intuitive picture of the benefits.
3. The paper is full of typos and erros:
* Lines 147-149: w in R^d, a in R^p, z=wa. The dimensions seem to mismatch. Inner-product of w and a is not typically written as wa.
* Line 157: backward propagation algorithm -> the backward propagation algorithm
* Line 162: Q_a represent -> Q_a represents
* Line 174: ‘s -> 's
* Line 176: does not works -> does not work
* Line 177:  Other latent-free optimizers, such as Bop (Helwegen et al., 2019b) and Bop2Order (Suarez-Ramirez et al., 2021) uses -> use

While I highly commend the authors for not using AI tools in their writing (at least not extensively), the paper needs to be thoroughly proofread before it's ready to be considered for publication.

---

### Decision · Program_Chairs · 2026-01-26

Reject